# Critical Analysis of the Snow Survey Network According to the Spatial Variability of Snow Water Equivalent (SWE) on Eastern Mainland Canada

**Yawu Noumonvi Sena \*, Karem Chokmani[ID], Erwan Gloaguen and Monique Bernier**

Institut national de la recherche scientifique-Eau Terre environnement (INRS-ETE), 490, rue de la Couronne, Québec, QC G1K 9A9, Canada; Karem.Chokmani@ete.inrs.ca (K.C.); erwan.gloaguen@ete.inrs.ca (E.G.); Monique.Bernier@ete.inrs.ca (M.B.)

\* Correspondence: noumonvi_yawu.sena@ete.inrs.ca; Tel.: +1-418-932-6775

**Abstract:** In Eastern Canada, the snow survey network is highly optimized at the operational scale. However, it is commonly accepted that the network is limited when it comes to studying the spatial variability of the snow water equivalent (SWE), which forms different spatial structures that are active at multiple scales—from local to regional. The main objective of this study was to conduct a critical analysis of the existing snow survey network, based on the spatial variability of the existing SWE structures. To do so, we must (1) assess the snow survey network's capacity to model spatial variability structures of SWE, and (2) study the spatial distribution based on the spatial variability structures of SWE. Initially, the snow survey network's capacity to model the spatial variability structures of the SWE was evaluated by a variogram analysis. Second, the spatial distribution of the snow survey network's data was analyzed through the Lorenz index curve and by measuring the spatial distribution using the Gini index. The results showed that, at a regional scale, the snow survey stations were evenly distributed within the spatial structures. However, at the local scale, the snow survey network was inadequate to model the spatial variability of SWE due to the reduced and uneven number of snow survey stations.

**Keywords:** snow water equivalent; spatial structures; variogram

## 1. Introduction

The snow survey network provides in situ data of the snow cover's physical parameters (density, depth, and snow water equivalent (SWE)). This snow survey network is optimized at the operational scale and provides real solutions to public safety issues (meteorology, hydrology, etc.) At the economic level, it also provides answers to practical problems related to hydroelectric reservoir management. However, in terms of climate analysis, the snow survey network is currently unevenly distributed and might not be entirely representative of the spatial variability of the snow cover, depending on the observation scale [1]. Indeed, the underlying processes (vegetation, atmospheric circulation, temperature, climate, relief, etc.), known in all points, generate spatial structures (or patches) of snow variability. A spatial structure is an entity or area in which the values of a given phenomenon (SWE in this case) exhibit a certain degree of homogeneity and a strong contrast with the values of surrounding entities [2].

In fact, at the scale considered, the uneven spatial distribution of the snow survey stations is also an obstacle to the quantitative analysis of spatial variability and the validation of spatial estimation of remote sensing algorithms. In such cases, the snow survey network's ability to reproduce the spatial variability of SWE is reduced. Studies on weather measurement station networks (rain, temperature,

etc.) [3,4] and hydroclimatic variables are abundant in the scientific literature. For example, in a study of the spatial distribution of weather stations in peninsular Spain, the concentration index (CI) and others correlated indices of daily rainfall were analyzed for a period covering 1951–2010. In this study, 32 first class meteorological stations were considered [5]. As is often the case for plant protection studies focused on vegetable crops, the study [5] was limited by the existing measurement network for the spatiotemporal modeling of agro-meteorological and environmental data [6]. These studies are generally carried out in response to a decreased density of survey station networks resulting from financial compressions by governments, changes in priorities and the needs of communities. For example, in the state of Illinois (USA), the station network was reduced from 358 to 189 stations between 1971 and 2001 [7].

The analysis of the Canadian network showed an increase in the number of survey stations from its beginnings in 1890 to 1930, which was followed by a 28% decrease in the 1940s because of the economic situation and the evolution of social priorities [8]. By the end of the 1950s, the network density had doubled to more than 3000 stations. Nonetheless, by the end of the 1980s, the Canadian network no longer met the World Meteorological Organization's standards regarding the minimum number of stations required in certain parts of the country [9]. Subsequent budget compressions led to shutting down many stations, and the network density decreased from 2608 to 1582 between 1984 and 1999 [8]. Few studies examined the capacity of the snow survey network to model the snow cover's spatial variability. There is, however, a study by Charbonneau et al. [10] that was carried out in the Lac Saint-Jean region, which contains recommendations for optimal mapping of the snow survey network. Redundant stations are identified by evaluating the type of precipitation recorded, the sampling interval, and the season. In order to optimize the accuracy of the estimates, various new station implementation scenarios are proposed. The capacity of a low-density network to capture the spatial variability of snow was also modeled at the regional scale [11]. By doing so, the stations of the province of Quebec were allocated to statistically homogeneous regions and the structures' functions were determined on the basis of the distance between the stations. At the local level, Tapsoba et al. [12] evaluated the degradation of the snow survey network of the Gatineau River watershed, using root mean squared errors (RMSE) of the kriging model (kriging with external drift), with altitude as the external drift. These snow station network analysis studies address dynamic network design strategies—a network's density type—to reduce the spatial interpolation variance of hydroclimatic variables. A network's ability to grasp the spatial variability of the phenomenon is barely addressed in these studies, where the data collected by the stations of a spatial structure are used to model the variability of SWE on other spatial structures, without any consideration for the limits and scale of the stations. This could induce modeling errors.

Measurement stations are designed to provide real data on the spatial variability of climatic elements. In this perspective, modeling the spatial variability of the snow cover (through the average annual maximum of the SWE) by using a functional approach, allows to objectively identify and explicitly delineate the boundaries of the different spatial structures of SWE, at a regional and local scale (10 km × 10 km and 300 m × 300 m, respectively) [13]. It has been shown that spatial variability structures of the snow cover differ according to the observation scale. Based on these delineated structures, a critical analysis of the existing station network can be conducted in relation to the spatial variability of SWE.

The main objective of the present study was to conduct a critical and objective analysis of the existing snow survey network, based on delineated spatial structures of the mean annual maximum SWE in the mainland part of Eastern Canada [13]. More specifically, (1) the current snow survey network's capacity to capture the spatial variability of the mean annual maximum SWE was evaluated, and (2) the spatial distribution of the snow stations in the delineated structures was analyzed. In the first part, the spatial variability of the mean annual maximum SWE was assessed through a variogram analysis in geographical areas with homogeneous spatial structures (in terms of mean

annual maximum SWE). In the second part, the Lorenz curve was used to characterize and measure the spatial distribution type of the stations in the delineated spatial structure.

## 2. Materials and Methods

### 2.1. Territory at Study

In Canada, a large part of the territory is covered by snow for a period of 5–8 months [14]. To the east of this vast territory (Quebec and adjacent Labrador), snow accumulation is significant at a continental scale, representing the second largest maxima with annual maximum snow accumulation averaging 200–300 mm of SWE [15]. Due to their geographical location and under the influence of the major atmospheric phenomena (ENSO, El Nino, maritime, and polar currents), snow accumulations in this area are characterized by a high variability over several decades, in Quebec and the adjacent Labrador [14]. In addition, the availability of various data (measurements of snow stations, density of the network, physiographic data, etc.) is one of the advantages of this case study. Indeed, the studied territory is partly located in the province of Quebec (43° to 65° North and from 55° to 82° West) and partly in Labrador, a continental region of Newfoundland and Labrador, two Canadian provinces (Figure 1). This huge territory presents a relief marked by relatively simple forms, which are—the Appalachians, Lowlands of Saint-Laurent River, and the Canadian Shield.

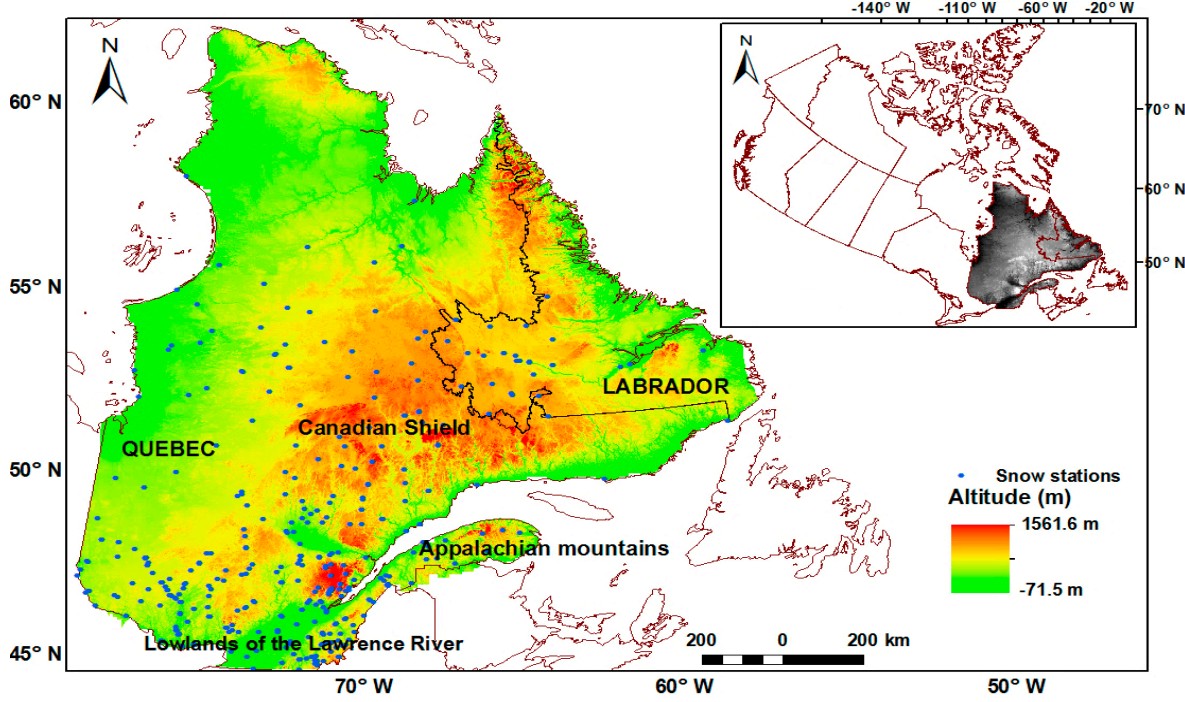

**Figure 1.** The studied territory.

The Appalachians are strongly eroded ancient mountains with narrow transverse valleys, which stretch from Newfoundland to the center of the State of Alabama (Southern United States).

Lowlands of the Lawrence River are the plains of the river abutting the Monteregian Hills (553 m).

The Canadian Shield has a flat relief, strewed with lakes; it is more hilly in the south where it includes the highlands of the Lawrence River (Trembling Mount 968 m).

Plant formation is strongly connected to climates and the area of study includes the northern temperate zones of plant formations (broad-leaved trees, sugar maple-hickory maple, etc.), the boreal zone (continuous boreal forest), and the Arctic zone (tundra, herbaceous, and shrubby, etc.). The climate of this vast territory undergoes strong variations, due to its relief, latitude, northern position, double exposure to the cold winds of the Hudson Bay, and the influence of cold oceanic streams on the

Labrador coast. Masses of polar air sweep the territory from the northwest through to the Hudson Bay, which is completely frozen in winter, while warmer, wetter air comes from the Southwest and Southeast. The annual precipitation (rain and snowfall) presents a contrast between the East and the West. Winter cyclones from marine origin mostly affect the eastern part of the area and leave significant snow accumulations.

　　　The snow cover is followed by a snow-station network that measures the physical parameters of the snow (density, thickness, and water equivalent of the snow). This unevenly distributed snow station network provides in situ measurements of the spatial variability of the snow cover (Figure 1).

*2.2. Snow Cover Data*

　　　The conventional protocol requires that the sampling site covers ten (10) measurement points, scattered over a length of 300 m. The measurement site must be representative of the variability of the surrounding snow cover and topography [16,17]. At each measurement site, snow depth was measured. A core sample was extracted, which allowed to evaluate the SWE and density of the snow pack. At each marker where the snow depth was greater than 25 cm, the weight of the core sample was measured. The SWE was obtained by subtracting the weight of the empty snow sampler from this weight. On the other hand, if the snow depth was less than 25 cm at one or more of the markers, the SWE was obtained after subtracting the weight of the empty snow sample bucket from the cumulative measurement of the cores of all markers. Values representative of the entire sampling site were obtained by calculating the mean of the 10 sampling sites. Measurements were carried out biweekly and were generally limited to the period from January to May [16].

　　　A ten-year observation period is the minimum necessary to cover cyclical atmospheric and oceanic events (solar cycle, El Niño-Southern oscillation, La Niña, etc.) that can influence the variability of the snow cover [18,19]. Based on the hypothesis that the snow phenomenon was stationary during a ten-year observation period, the sub-periods were not considered. Therefore, out of the 426 snow survey stations constituting the network of the studied territory (Figure 1), we considered the 367 stations for which historical data was available for at least the last 10 years. Following this hypothesis, 367 stations containing 10 years and more of data were included in this study, including data from 193 stations of the province of Quebec's Ministère du Développement Durable, Environnement et Lutte contre les Changements Climatiques [16], 19 stations belonging to RioTinto, 76 to Hydro-Québec, and 79 provided to Environment and Climate Change Canada. For each retained station, the mean annual maximum SWE was calculated. Thereafter, statistical descriptors of central tendency (mean of annual maxima) and dispersion (standard deviation, interquartile interval, median) of the mean annual maximum SWE were calculated.

*2.3. Spatial Variability Structures of the Snow Cover*

　　　Sena et al. [13] conducted a multi-scale analysis of the spatial variability of the SWE in Eastern Canada using a functional approach. This was achieved under the assumption that the snow phenomenon was stationary over the entire observation period. First, the different spatial variability structures of the mean annual maximum of the SWE were visually identified, according to their geographical positions (latitude and longitude). They were then quantitatively selected using the spatial association index, based on the similarity of the values collected at each station. Using these two methods we were able to demonstrate that the SWE was not stationary across the study area, neither in terms of its mean nor its variance. However, spatial structure limits remain subjective, which justifies the use of spatial segmentation. Subsequently, the spatial segmentation algorithm was applied to explicitly delineate the boundaries of the structures of spatial variability of the SWE. This algorithm was applied to the physiographic metavariables obtained using a canonical correlation analysis. Physiographic factors are often optimized to explain target variables, using multivariate statistics such as principal component analysis (PCA) or the canonical correlation analysis (CCA) [20,21]. The CCA is a statistical multivariate analysis tool that allows us to describe the relationship of dependence existing

between two sets of random variables [22]. It also allows to determine linear combinations pairs for each set of variables (canonical variables), so that the correlation between the canonical variables of a pair is maximized and the correlation between the variables of different pairs is zero. This method has the advantages of providing physiographic meta-variables (U) optimized by the SWE data that are, known at any point in space. Such physiographic metavariables provide information on the spatial variability of snow, at each scale considered, since they are correlated with the SWE. Finally, the results of the spatial segmentation were validated, by comparing snow data from the adjacent geographical areas, using the Kruskal–Wallis non-parametric statistical test. Thus, at the regional scale (10 km × 10 km), the multi-resolution spatial segmentation algorithm had made it possible to identify six geographical areas with homogeneous spatial structures, in terms of SWE (Figure 2a). These zones were defined by the underlying processes at the regional scale, which are generated by physiographic factors, their geographic position (latitude and longitude), the relief, and their distance from the ocean. These geographical areas are similar to the layout of landforms (altitude) and climate classes. At the local scale (300 m × 300 m), spatial segmentation has allowed the identification of several local geographical areas. The identified local structures of the spatial variability of the SWE demonstrate the dominant role of the physiographic factors (slope, curvature, etc.) (Figure 2b) in the maintenance and redistribution of snow cover (for more information refer to Sena et al. [13]).

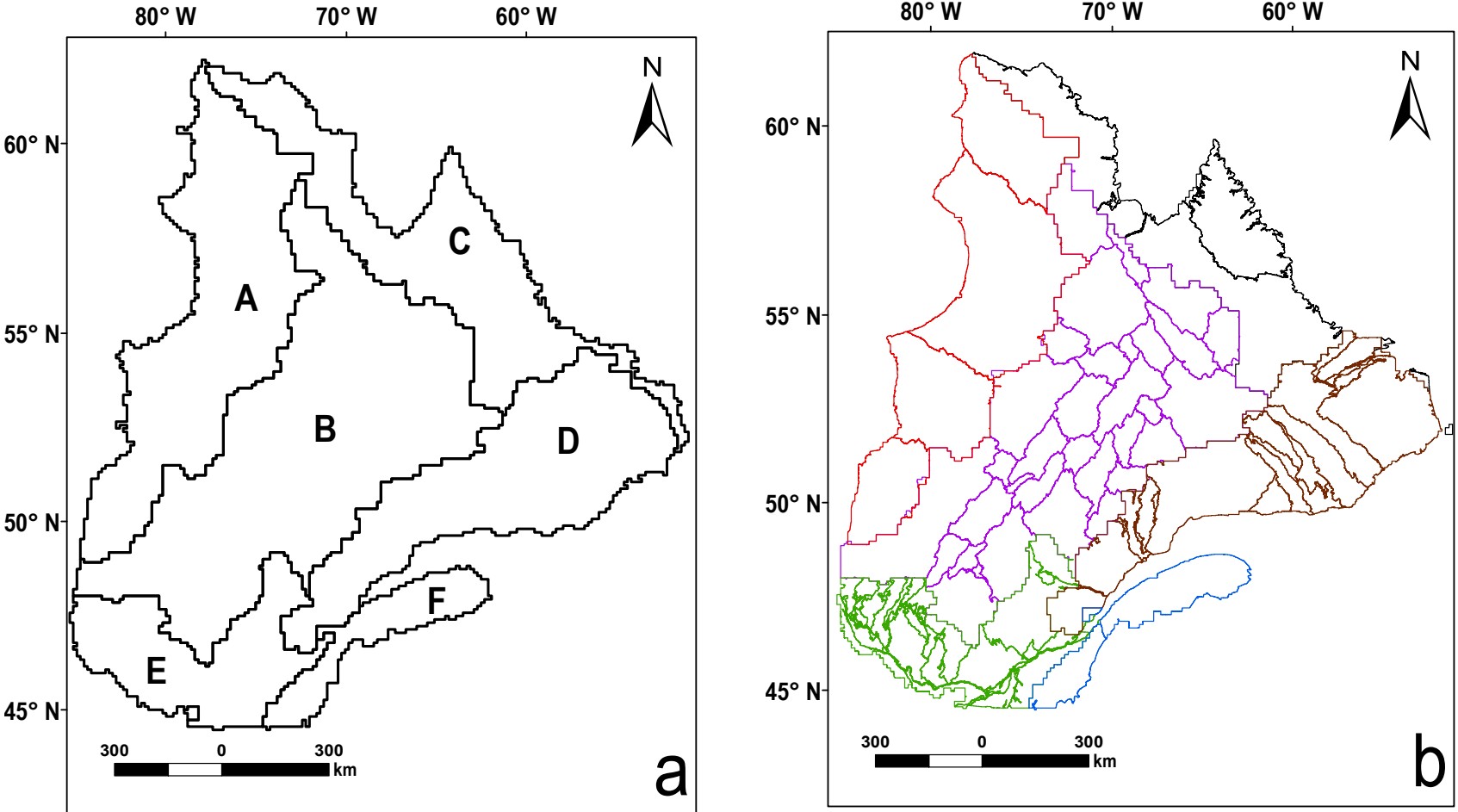

**Figure 2.** Zones and geographical units of homogeneous snow water equivalent (SWE) spatial structures obtained by the functional method at regional scale (**a**) and local scale (**b**).

*2.4. Methods*

2.4.1. Analysis of the Spatial Structures

The ability of the snow survey network to identify the SWE variability was assessed by a principal component analysis, the estimation of the function of SWE structures, and optimal interpolation, to identify statistically homogeneous station groups [11,23]. Alternative methods used to evaluate the measurement network included the use of root mean squared errors of the kriging and variogram analyses [12,24,25]. Sometimes, data collected from the stations of a structure are taken into account to model the spatial variability of the SWE on other structures, without taking into account their limits or the scale considered. This might lead to modeling errors in the variability of the SWE. To avoid this, opting for a particular approach, including the variogram analysis of each delimited structure, has the advantages of taking the spatial limits into account and reducing the modeling errors of the variability of SWE, allowing more effective modeling of the spatial variability of the SWE. This approach makes it possible to define a strategy for the maintenance of the most representative and informative distribution of SWE survey sites. Such a strategy can be defined by using the variogram analysis of the mean annual maximum of SWE, leading to the quantification of the spatial structure of regionalized variables [26,27]. The variogram measures the dissimilarity between the values separated by a distance h [27]. It, thus, allows us to determine whether a phenomenon's distribution is regionalized, random, or periodic [26]. The experimental variogram is generally adjusted with an analytical function, as follows:

$$\gamma = \frac{1}{2N(h)} \sum_{i=1}^{N(h)} [Z(x_i) - Z(x_i + h)]^2 \tag{1}$$

with the variables defined as follows:

- $-\gamma(h)$ is half of the difference of the mean square deviation between SWE dataset pairs at the stations $X_i, \dots X_n$;
- $-N(h)$ $(h)$ is the number of station pairs separated by $h$ (translation vector);
- $-Z$ is the random function (the mean annual maximum SWE);
- $-Xi = (X_{i1}, \dots X_{id})$ is the coordinate of the $i$th station; $Xi$ stations belong to a domain D;
- $-h$ is the distance vector between two arbitrary stations $Xi$ and $Xj$, defined by:

$$h\mathrm{ij} = x_i, \dots \ x_j = (x_{i1}\text{-}x_{ij}, \dots , x_{id}\text{-}x_{jd}). \tag{2}$$

The variogram is characterized by its behavior at the origin, also known as the nugget effect ($C_0$; when $h = 0$), its sill ($C$), representing the variance of the random function and the range, the distance after which two points of the random function are not correlated anymore. Its behavior at the origin, corresponds to the intrinsic variance of the random function at distance $h = 0$ (or noise) or a small-scale variation of the phenomenon whose spatial structure is smaller than the sampling area. For more information about the variogram, refer to [26] and [27]. Sometimes $C$ and $C_0$ are merged in a single index by considering the following ratio:

$$\beta = C/(C_0 + C) \tag{3}$$

where

- ✓ If β is high, the nugget effect is small (<0.5), which translates into increased spatial variability due to the distance between the stations. In which case, $C_0 < 10\% \times C$.
- ✓ If $\beta$ is low, the nugget effect is high (>0.5), and the spatial variability can be explained by the nugget effect.

Several works have previously analyzed and modelled the spatial variability of climate elements [24,25]. For example, the variogram analysis was used to describe the variability of snow measurements and to define the spatial structures of the snow pack on slopes [28]. The spatial structures of snow-depth on the slopes were studied using an experimental semi-variogram, on the residuals of the linear regression on snow-depth and coordinates of the fracture initiation and propagation properties of the snowpack. The latter largely control the avalanche formation process [28].

The evaluation of the snow survey network's capacity to provide reliable spatial structures, requires a sufficiently large sample size (at least 50 to 100 stations) [29]. The reduced number of measurements or the density of samples increases the nugget effect (the microstructures of the phenomenon). Indeed, the increase in measurement density generally improves the reliability of experimental semi-variograms [30–32]. In this work, the variogram analysis was carried out within zones at a regional scale, because the number of sampling stations was sufficient (Table 1). Indeed, at a regional scale, the number of SWE stations was three in zone C, located to the north against 128 stations in zone E in the south (Table 1). For this purpose, the variogram analysis was carried out only in the B, D, E, and F zones. Moreover, some locally homogeneous areas in terms of SWE are free or less of snow monitoring sites (Table 1).

**Table 1.** Homogeneous zones at the regional and local scale and SWE statistical values.

| Spatial Variability | | | | | |
|---|---|---|---|---|---|
| Regional Scale (10 km × 10 km) | | | | | Local Scale (300 m × 300 m) |
| Homogeneous Zone | Number of SWE Stations | Mean * | Min ** | Max *** | Number of Local Homogeneous Zones |
| A | 18 | 183.08 | 0 | 240.1 | 4 |
| B | 103 | 146.6 | 0 | 398.17 | 33 |
| C | 3 | 262.8 | 0 | 337.8 | 3 |
| D | 50 | 248.06 | 0 | 515.02 | 23 |
| E | 128 | 165.04 | 0 | 398.05 | 46 |
| F | 58 | 208.27 | 98.25 | 350.64 | 2 |

* number of SWE stations per regional homogeneous zone; ** minimum of SWE per regionally homogeneous zone; *** maximum of SWE per regionally homogeneous zone.

2.4.2. Study of the Spatial Distribution of Snow Survey Stations

The spatial distribution of the snow survey stations represents the space covered by the stations. This concept is often studied in the statistics applied to economic, demographic, and health studies, where the notion of spatial distribution is more commonly addressed [33,34]. Several dimension indices such as the Gini index (GI), also known as the Gini coefficient or the Theil index (TI) are used in the study of precipitation concentration [5,35] and in the analysis of rainfall characteristics [5,31,36,37]. The TI is related to a particular generalized entropy index and does not depend on any additional parameter [38]. The Lorenz curve is established and allows the assessment of the unevenness of the distribution of precipitations. Derived from the GI, the Lorenz curve only uses the empirical curve, representing a sample of the rainfall (observed values) [5]. In addition, by applying the GI to the snow measuring stations of each defined structure, it is possible to develop relocation strategies for the measuring stations.

The adopted approach to study the spatial distribution of snow survey stations is based on the analysis of the Lorenz index and the measurement of the distribution's evenness, using the Gini index. Measuring spatial inequalities amounts to a characterization of the distribution of a statistical series (of stations, resources, etc.). Inequality measurement involves all dispersion and concentration indicators in a statistical series [39]. The Gini index is a statistical measure to account only for the unequal

spatial distribution of stations. Inequality is measured using a synthetic indicator that summarizes the dispersion of the spatial distribution of stations and can be represented as a graph, such as the Lorenz curve. The Gini index is an indicator of the type of normative measure [40]. The Lorenz curve is a representation of the cumulative share of snow survey stations (*x*-axis) compared to the cumulative proportion of the annual averages of the mean annual maximum SWE (*y*-axis) (Figure 3).

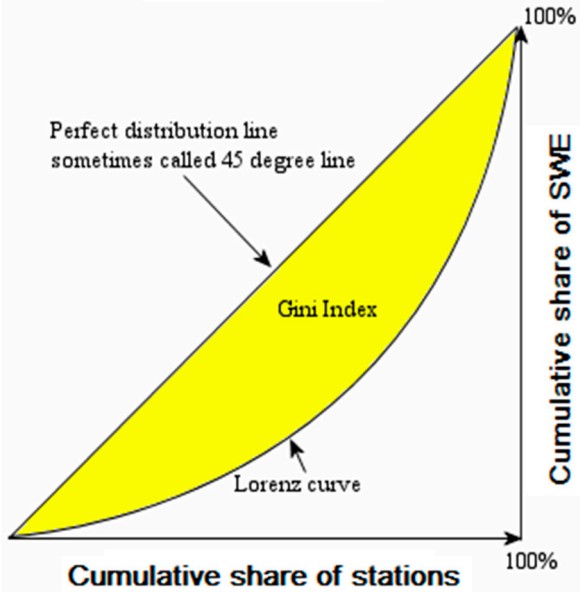

**Figure 3.** Graphical representation of the Gini index.

The bisector represents an even distribution of the stations. As the percentage of nearby stations increases in a small area, i.e., their distribution is not proportional to the space, the deviation of the Lorenz index towards low values will increase. This uneven distribution is measured by the Gini index, which is calculated by reporting the surface of the triangle formed by the bisector, to that of the area delineated by the curve of the mean annual maximum SWE. A Gini coefficient close to 1 indicates that the majority of the stations are unevenly distributed, and a Gini coefficient close to 0 indicates that the stations are relatively evenly distributed in space.

On the other hand, the GI adapted to SWE is obtained by using the formula developed by Brown [41]:

$$GI = 1 - \sum_{i=1}^{n} (X_i + X_{i-1})(Y_i + Y_{i+1}) \tag{4}$$

with n representing the number of stations, X being the sum of the number of stations (cumulative share of stations), and Y being the sum of the mean annual maximum SWE values (cumulative share of SWE).

## 3. Results

### 3.1. Analysis of Spatial Structures

Spatial structure analysis was carried out, based on the data collected from stations in geographic areas with homogeneous spatial structures of mean annual maximum SWE (B, D, E, and F) (Table 2). The method applied to zones A and C was very limited, due to the reduced number of stations (18 and 3, respectively).

**Table 2.** Parameters of the variograms of the zones B, D, E, and F, at the regional scale.

| Zone | Nugget Effect ($C_0$) ($mm^2$) | Range (km) | Variance C = $\gamma(h)$ ($mm^2$) | $\beta = C/(C_0 + C)$ |
|---|---|---|---|---|
| B | 200 | 600 | 2890 | 0.93 |
| D | 950 | 210 | 11,000 | 0.92 |
| E | 154 | 174 | 2100 | 0.93 |
| F | 300 | 360 | 3300 | 0.91 |

In zone B, the variogram analysis indicated that the stations were correlated with each other, up to a distance of 600 km and show a range (sill) of $\gamma(h_B)$ = 2890 $mm^2$ (Figure 4, zone B). The index values ($\beta_B$) showed that the spatial structure was mainly due to the distance between the stations. A similar observation could be made in zone D, where values of $\beta_D$ and $C_0 < 10\%$ C showed that the mean annual maximum SWE's spatial variability structure was a function of the distance between stations. Indeed, the variogram of zone D (Figure 4, zone D) had a spatial structure (sill) of $\gamma(h_D)$ = 11,000 $mm^2$ up to a distance of 210 km and showed the highest nugget effect (950 $mm^2$) up to a distance of 9 km, as observed in this study. The short distance between the higher values of the maximum annual average SWE of the stations in Piedmont (Raoul-Blanchard and Veyrier, for example) and the lower values observed in the Saguenay and Manicouagan basins, could be explained by the high values of the pinch effect. Zone D crosses a large part of the Laurentians (Canadian Shield) and a low coastal valley. The heavy snowfalls of the Laurentian highlands were located at a very short distance from the high values located along the coast. This might explain the discontinuity of the regionalized variable (average annual maximum of the SWE). At about 210 km, the SWE values of the two snow stations were no longer similar at all, on average, and they were no longer linearly linked (linear covariance). Beyond this, the variance ($\gamma(h_D)$ = C + $C_0$) of SWE was reached.

Zone E covers a large part of the St. Lawrence Lowlands and has the highest measurement station density in the study area. This high density of stations (Table 1) explains the low value of the nugget effect observed in this area. The nugget effect $C_0$ = 154 $mm^2$ was the lowest value observed in a microstructure, among the microstructures identified within all zones. As mentioned above, the values of $\beta_E$ and $C_0 < 10\%$ C in the zone E showed that the SWE had a variance of $\gamma(h_E)$ = 2100 $mm^2$), which was a function of the distance between the pairs of the snow measurement stations pairs (Figure 4, zone E). In zone E, the value of the variable $\gamma(h_E)$ (C + $C_0$) was lower, compared to zones D and B. The maximum value of SWE was equal to the maximum value found in zone B (Table 1), but the variance ($\gamma(h_E)$) was low, compared to the one of zone B. This lower value of variance in zone E could be explained by the geographic location (lower latitude) and the lowlands of the St. Lawrence River where snow accumulations were low under lower latitudes.

In zone F, the SWE had a variance of 3300 $mm^2$ at 360 km. The small structure (nugget effect, $C_0$) was greater than that of zone B (Figure 4, zone F). The variance ($\gamma(h_F)$ = 3300 $mm^2$) was the second highest, probably because of the influence of the relief, the maritime influence and the proximity to the plains of the lowlands of the St. Lawrence River. It had a great influence on the zone's spatial structure (C), since $C_0$ was slightly less than 10% C (Table 2). However, the value of the index $\beta_F$ (Table 2) indicated that the spatial variability structure of the mean annual maximum of SWE in zone F was based on the distance between the pairs of snow measuring stations. This was probably due to the high relief (Appalachian Mountains) in the northern part, which has fewer measuring stations than that in the southern part, and has a high human concentration.

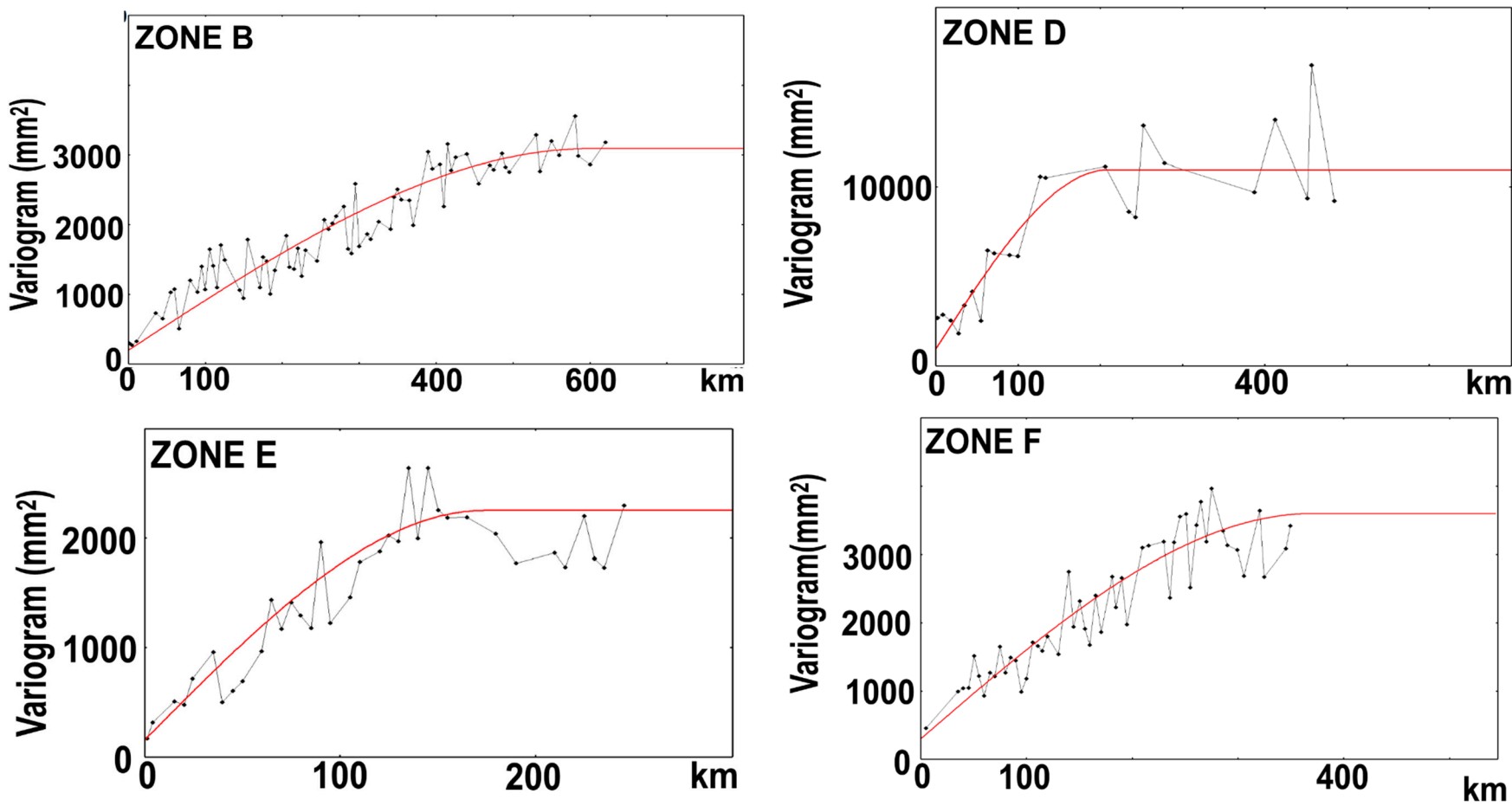

**Figure 4.** SWE variograms in zones B, D, E, and F. The experimental variograms (black) and the adjustment models (red).

In all geographical areas (B, D, E, and F), the variogram analysis provided a detailed assessment of the spatial structure of the mean annual maximum SWE. The snow survey stations model the spatial structure of the mean annual maximum of SWE, for distances specific to each area. The structures in the areas have different shapes, variances, and small structural differences. Since the stations unevenly cover the areas, it was necessary to assess the spatial distribution of the stations in order to determine whether the snow survey network evenly covered the delineated structures.

### 3.2. Spatial Distribution of the Snow Survey Stations in Areas with Homogeneous Spatial Structures

At the regional scale, the snow survey stations within the homogeneous spatial structures (in terms of the mean annual maximum SWE) were evenly distributed. The Lorenz curves in the six zones were nearly merged with the bisector. This is shown in Figure 5, where the Gini coefficient values are near zero (0.01–0.02).

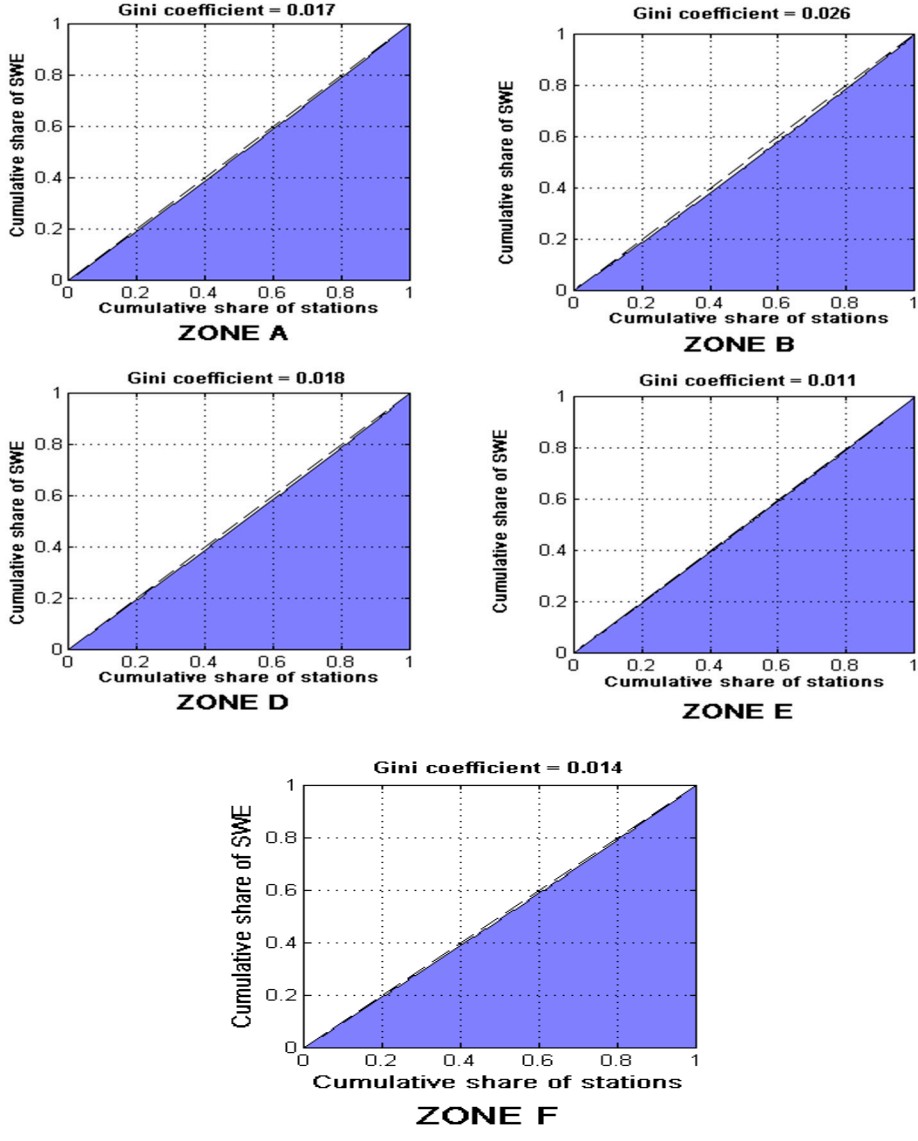

**Figure 5.** Lorenz curves and Gini coefficient of zones A, B, D, E, and F at the regional scale.

At the local scale, the spatial distribution of snow survey stations was not the same in all delimited geographic units. In unit 2 of zone A, the Lorenz index curve showed that approximately 90% of the stations provided only 72% of the mean annual maximum data (Figure 6). This slight spatial unevenness of the stations in unit 2 (Gini coefficient = 0.17) showed that the number of stations

decreased with increasing altitude, to reach a single station (Inukjuak: latitude 58.45° N; longitude 78.11° W), on the northern coastline of the study territory. On the other hand, in unit 3 (under 55° N), the stations had an even spatial distribution with a Gini coefficient of zero.

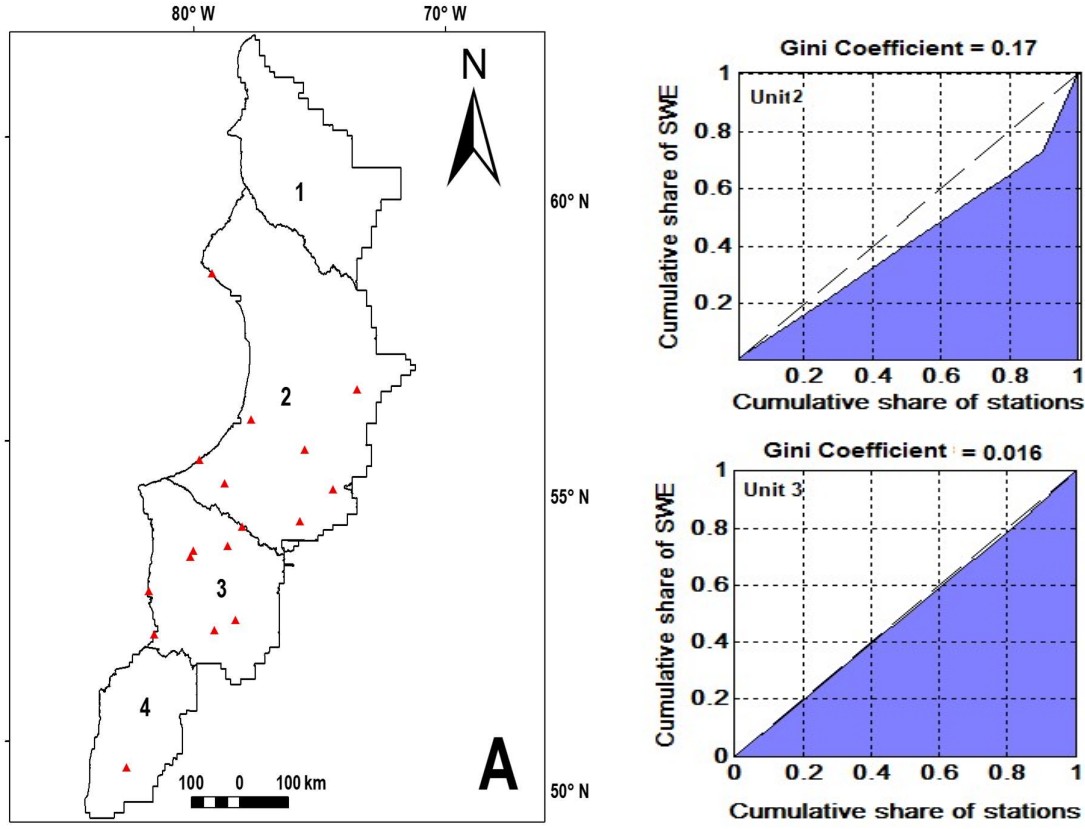

**Figure 6.** Lorenz curves and values of spatial distribution unevenness of stations in units 2 and 3 at the local scale of zone A (units 1 and 4 lacked sufficient number of stations to undertake the analysis).

Spatial unevenness varied in the units of zone B (Figure 7). In unit 1, the snow cover was monitored by a network of evenly distributed snow stations (Gini coefficient = 0) around the main lakes of Churchill Falls' hydroelectric station. The stations of unit 10 in the southern part of zone B were distributed similar to those of unit 1, with a slight station grouping in the southern center. West of zone B, in unit 2, 62% of the mean annual maximum SWE data were derived from 82% of the stations, which were located along the central Great Lakes. In the fifth unit of zone B, in the South, on the tributaries of the Manicouagan Reserve and the foothills of the Veyrier and Groulx Mountains, 70% of the snow stations contributed 35% of the snow data. As a result, the entire northern part of unit 5 was devoid of stations, resulting in a Gini coefficient of 0.34. Such an uneven distribution could be observed in unit 6, where the stations that were distributed east of the mountain foothills (Tesmicamie 554 m and Helon 467 m) and at the edge of the Northern lakes (38% of the stations) provided only 15% of the mean annual maximum SWE data. The central part of unit 6 was also not well-covered, which was confirmed by a Gini coefficient of 0.22. The spatial distribution of the stations in unit 7 was even less scattered, where the stations were aligned on the North–South axis, resulting in a Gini coefficient of 0.3. This uneven spatial distribution was confirmed, with 49% of the stations providing 19% of the data of the mean annual maximum SWE. A similar Gini coefficient value (0.27) was observed in unit 9, where 35% of the stations provided less than 10% of the mean annual maximum SWE data.

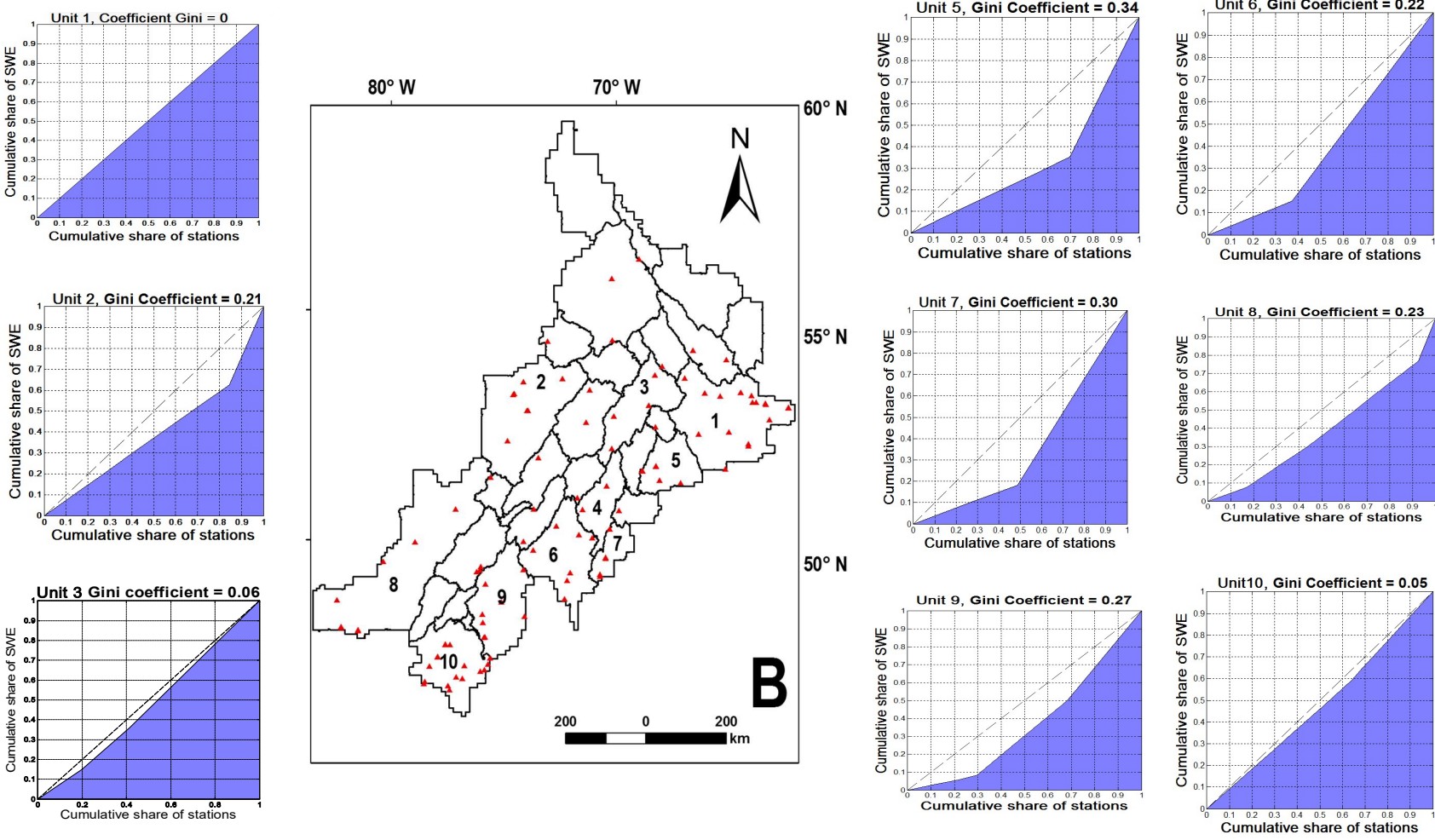

**Figure 7.** Lorenz curves and values of the spatial distribution unevenness of stations in the units of zone B, at the local scale.

The stations were concentrated around the southernmost water bodies and along the Ashuapmushuan wildlife reserve. In the eighth unit of zone B, the spatial distribution of the stations showed that 77% of the data were provided by 92% of the stations. The stations were located in the highly populated areas of the south, and along the tributaries of the main lakes in the north and the center of unit 8.

The stations of zone D's unit 1 were unevenly distributed (Gini coefficient = 0.34) with 39% of the mean annual maximum SWE data derived from 72% of the snow survey stations (Figure 8). As was the case for the other units, these stations were located around the mouth of the lakes and close to the populated areas. Similarly, in unit 2 which was in the southern part of Lac-Saint-Jean, the presence of national parks (Hautes-Georges-de-la-Rivière-Malbaie, Jacques-Cartier, etc.) and the Laurentides wildlife reserve explained the even distribution of stations. The snow cover at the foothills of the Grand-Jardins mountains (Mont Raoul-Blanchard, Mont du Lac des Cygnes, Monts Érables, etc.) was monitored by equally distributed stations (Gini coefficient = 0.1). To the North, the Saguenay River was monitored by a series of stations (92%), which provided 83% of the mean annual maximum SWE data.

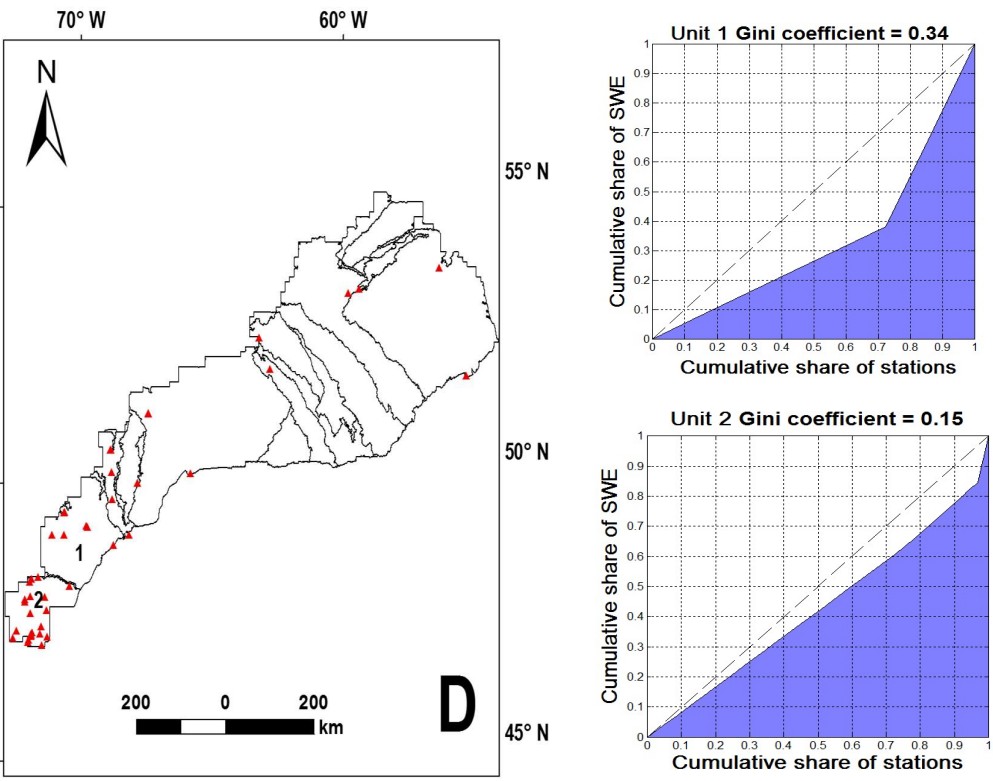

**Figure 8.** Lorenz curves and values of spatial distribution unevenness of stations in the units of zone B, at the local scale.

Overall, the snow survey stations in zone E were unevenly distributed, with the exception of unit 1. This was illustrated by the Lorenz index curves shown in Figure 9. The even distribution of the stations of unit 1, from the Cabonga Lake to the south of the Baskatong Reservoir, was due to the presence of the La Verendrye Wildlife Reserve. On the other hand, in unit 2, which was delimited to the North by Lac-Saint-Jean, 68% of the stations provided 37% of the data, which corresponded to a Gini coefficient of 0.4. This was due to a lack of stations in the South. This unit encompassed a portion of the Laurentides Wildlife Reserve, which explained the clustering of stations in the North, around the tributaries of the lakes.

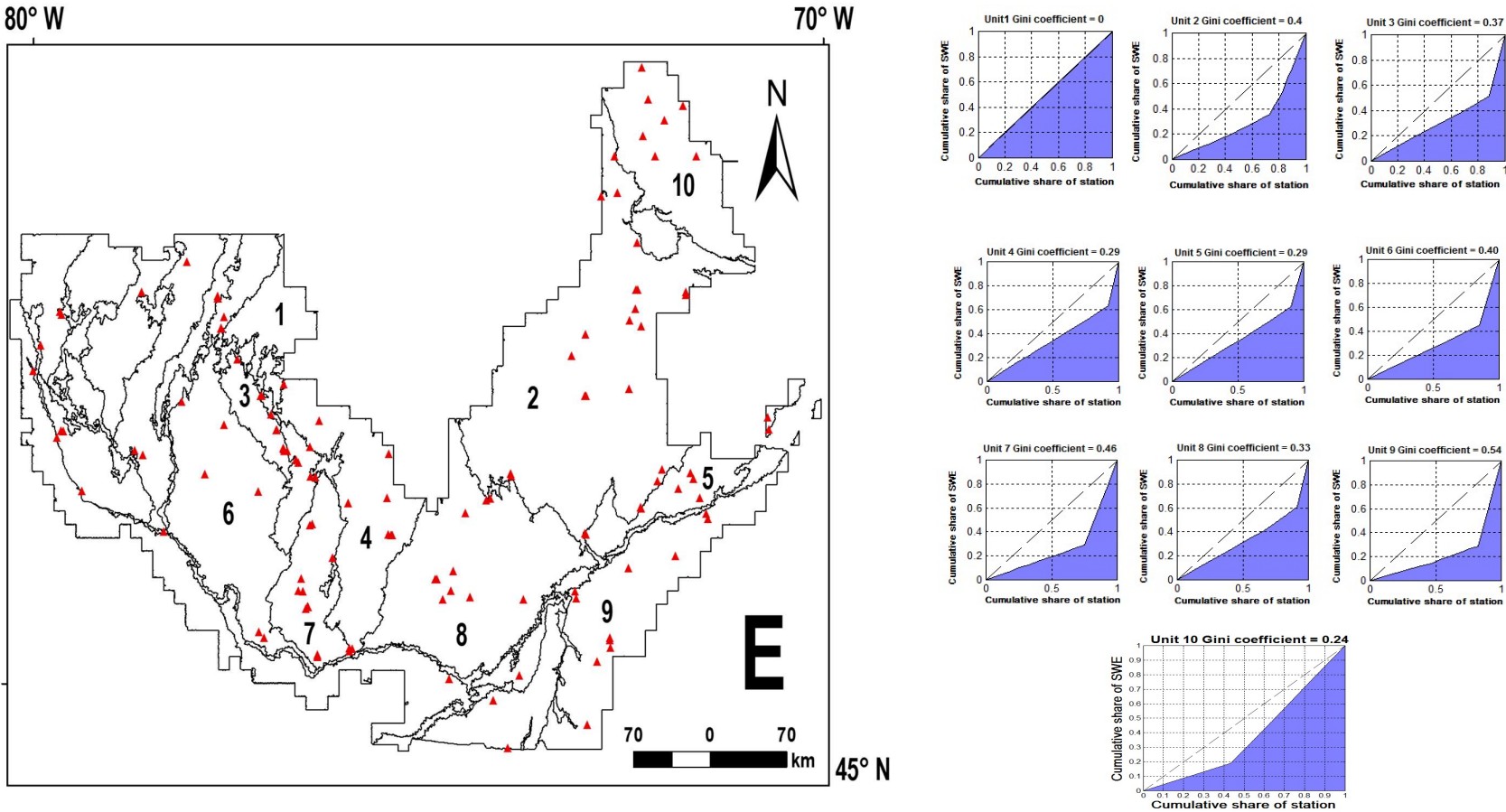

**Figure 9.** Lorenz curves and values of spatial distribution unevenness of stations in the units of zone E, at the local scale.

In the case of unit 3, which covered a large part of the La Verendrye Wildlife Reserve, the stations were located along the road axis, and the corresponding Gini coefficient was 0.37. In this unit, almost 57% of the mean annual maximum SWE data were provided by 82% of the stations. Similarly, the stations (92%) of unit 4 were grouped around the sites of Mount Laurier, providing 61% of the mean annual maximum SWE data.

This uneven distribution of stations (Gini coefficient = 0.29) indicated inadequate monitoring in the southern part of the unit, all the way down to the Ottawa River. A similarly uneven distribution of the stations was observed in unit 5, where the measurement stations were concentrated around the city of Québec (airport and Valcartier). In the western part of zone E, unit 6 stretched from the La Verendrye wildlife reserve to the Ottawa River in the south. The stations are part of the Pontiac controlled harvesting zones (ZEC), located southeast of the cities of the great plain of the Ottawa River. This uneven station distribution corresponded to a Gini coefficient of 0.4, where 41% of the mean annual maximum SWE was collected by 81% of the stations. A similar value of unevenness was observed in unit 7, which included the Gatineau River watershed. Stations (75%) that were set up along the river and the tributaries of the lakes north of the unit, provided 37% of the collected data. The small number of snow survey stations in the basin were previously highlighted in [12]. In unit 8, the stations (92%) grouped in the vicinity of Mont Tremblant and the main road axis only provided 60% of the mean annual maximum SWE data. This uneven distribution (Gini coefficient = 0.3) was characterized by stations in the center and in the Saint-Michel-des-Saints agglomeration of the Lac Taureau regional park.

The clustering of stations around the urban areas was a distinct feature of unit 9, in which the distribution was highly uneven (Gini coefficient = 0.5). In fact, the stations (81%) located in the housing areas of the St. Lawrence River plain only provided 22% of the mean annual maximum SWE data. North of zone E was unit 10, in which stations (41%) located on the foothills of the Témiscamie Mountains provided 20% of the mean annual maximum SWE data. This uneven distribution (Gini coefficient = 0.2) showed a lack of stations in the south and could be explained by the importance of monitoring the snow cover in areas of economic interest (winter sports).

In zone F, the stations of unit 1 were scattered along the main roads, with a marked lack of stations in the central area (Figure 10). As a result, almost 29% of the mean annual maximum SWE data were provided by 82% of the stations. The stations were more evenly distributed (Gini coefficient = 0.1) in the southern part of unit 2, where 37% of the stations provided 25% of the mean annual maximum SWE data. It is an area of high human density with a dense network of stations that appropriately covers the area.

In short, it should be noted that the snow survey stations in the study area unevenly covered the area at the local scale. There were high concentrations of stations in the sectors of economic interest, urban agglomerations, along road axes, etc. These stations met the local needs (public safety, water resources management, etc.) and were highly operational [14].

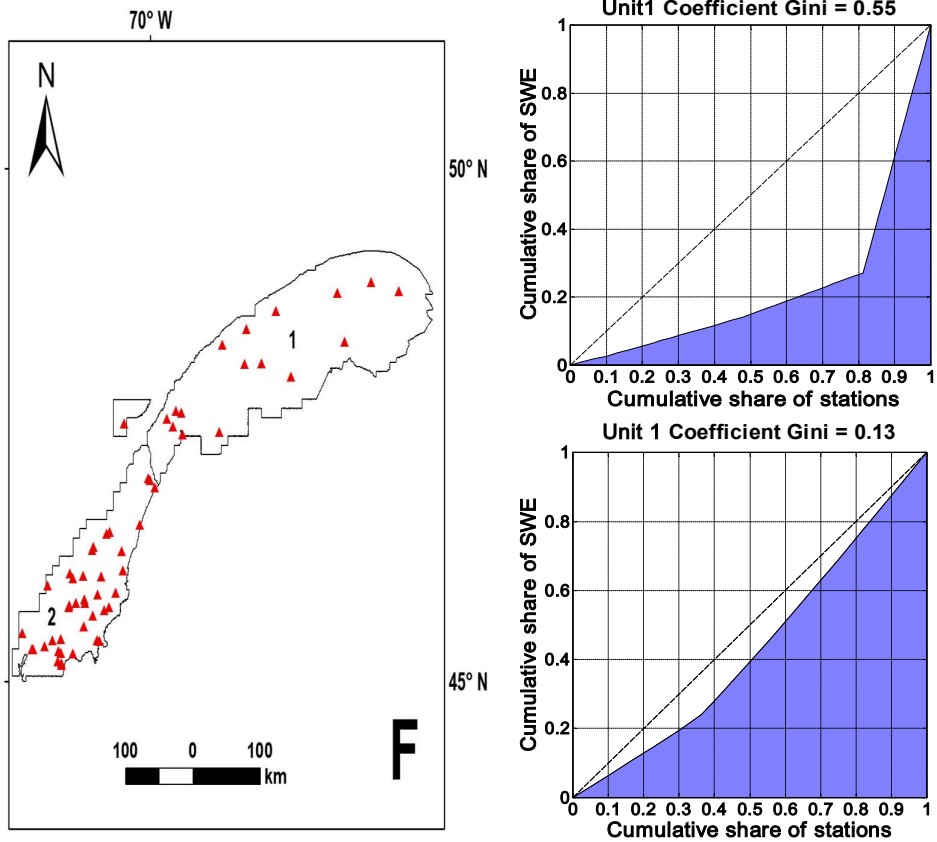

**Figure 10.** Lorenz curves and values of the spatial distribution unevenness of stations in the units of zone F, at the local scale.

## 4. Conclusions

The aim of this article was to critically analyze snow survey station networks on the delimited homogeneous spatial structures of the mean annual maximum SWE data [13]. This goal was first achieved by evaluating the ability of the measurement network to model the spatial structure of the mean annual maximum SWE, using a variogram analysis. In a second phase, the spatial distribution of the stations was assessed by analyzing the Lorenz curves and by measuring this spatial distribution with the Gini index.

At the regional scale, the variogram analysis allowed to characterize the delineated structures of spatial variability of the mean annual maximum SWE in the areas of study. The variogram analysis was conducted in the four regions (B, D, E, and F), out of six that had at least 50 snow survey stations. Zones A and C (14 and 3 stations, respectively) did not have enough stations for a variogram analysis to be carried out. The analysis of the ability of the snow sites survey stations to provide the spatial variability of the mean annual maximum of the SWE showed that the value of the mathematical descriptors of the structural elements (variances, ranges, small-scale variations) varied from one region to another, at the regional scale. In all areas, at small distance, the nugget effect was identified. The high variability of the selected structures was due to the high values of the SWE in the mountainous zones located near low valleys (example of zone D) or the high density of the measuring network (zone E). The strong variance of zone D ($\gamma(h_D)$ = 11,000 mm$^2$) showed the role of relief in the spatial variability of the SWE. Additionally, the high variance ($\gamma(h_F)$ = 3300 mm$^2$) (second highest) was most likely explained by the relief, the maritime influence and the proximity of the costal lowlands. However, in all zones, index $\beta$ indicated that the variance of the mean annual maximum of the SWE was due to the distance between the pairs of stations on each zones.

The results of the spatial distribution analysis showed an acceptable and even distribution of stations at the regional level. Outside of zone C, in the north, the spatial distribution of stations in the rest of the study area (86% of the total area) was used to characterize spatial variability at the regional level.

At the local level, no delineated geographic unit had a sufficient number of sites to carry out the variogram analysis. Certain units were completely devoid of stations. Twenty-three percent of the covered territory did not have a local station. In the geographic units where the stations were present, the variogram analysis could not be performed because the distance between the stations did not allow us to capture the spatial variability over a short distance. In certain units, the snow survey station network was inadequate to study the spatial distribution and to capture the spatial variability of the mean annual maximum SWE, at the local scale. In other geographic units, it would be necessary to install or move several stations to optimize the monitoring. Very often, stations were grouped around the areas of interest (economic activities, roads, wildlife reserves, etc.).

This analysis showed that the snow survey station network provided only partial data regarding the spatial variability of the snow cover, over the entire study area. For this purpose, spatial estimation methods such as geostatistics [42], neural networks [43], physical models [44], and hybrid models were proposed to estimate the snow cover outside the snow station network. Remote sensing (optics, passive or active microwave, etc.) was another alternative source of indirect data on the snow cover. However, the validation of algorithms based on in situ data must take into account the limits of the spatial variability structures of the snow cover, in order to limit the estimation errors at the scale considered.

However, despite the fact that the data collected from the stations were incomplete and that their spatial distribution was uneven, they remained the main sources of in situ continuous snow cover data on which the spatial models were based.

This study draws a critical portrait of an existing snow survey network based on delimited structures. As it shows that the spatial variability structures of snow cover differ according to the scale of observation and is based on the delineated structures, a critical analysis of the current station networks could be conducted. This helped assess the ability of snow survey station networks, to provide information on the spatial variability of SWE.

The main limitation of this method was the size of the sample (number of stations) and its representativeness of the phenomena. Indeed, the number of points used in the variogram analysis was an important factor in reducing the uncertainties and large confidence intervals. At the regional scale, the minimum number of stations necessary in some areas (50 and 58, respectively, in zones D and F) could lead to uncertainty and errors in the variogram analysis. Additionally, at the local scale, the limited number of stations available in the delimited structures did not allow for variogram analyses to be conducted. In a future study, the number of stations in each area could be increased by changing the observation scale of the spatial variability of SWE. Additionally, this method could be applied to other physical parameters of snow (density, height) or climate (rain, temperature, etc.), and other variables of interest (annual minimum, monthly maximum, monthly average, etc.). This study might lead to further optimization analyses of the snow or weather survey network by relocating snow survey stations.

**Author Contributions:** This article included the result of the second objective of Sena's thesis, co-supervised by K.C., E.G., and M.B. Sena analyzed the data; E.G. supervised the geostatic approach; K.C. contributed to data collection and provided global supervision; and M.B. supervised the analysis of SWE spatial variability.

**Funding:** This research was funded by Institut national de la recherche scientifique Centre Eau-Terre-Environnement Québec, QC, Canada.

**Acknowledgments:** The authors thank the Ministère du Développement Durable, de l'Environnement de la Lutte contre les Changements Climatiques [16], Environment Canada, Rio Tinto and Hydro-Québec for providing the snow water equivalent data.

**Conflicts of Interest:** The authors declare no conflict of interest.

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
