# Peer review of "Critical Analysis of the Snow Survey Network According to the Spatial Variability of Snow Water Equivalent (SWE) on Eastern Mainland Canada"

_hydrology, doi:10.3390/hydrology6020055_

Round 1

Reviewer 1 Report

The introduction, problematic and objectives must be rewritten in a more logical and  coherent way. At present it is very general and often vague. A few concepts, such as 'delineated structures', 'spatial structures', 'model' are used without explaining what they mean. Altought I understand the objectives of the study, the introduction and problematic currently do not correctly place the research objectives in the context of existing knowledge about spatial variability of SWE and its measurement.

METHODS

The description of data and methods is not sufficiently detailled or well explained.

- It is not clear if the 367 sites are all snow survey networks with similar measurement protocols

- number of station per regional and local zone. The table on page 7 has no number and is not cited in the text. It seems to also contain innacuracies (mean SWE vs mean number of SWE stations?). I question the arguments provided for 'having a sufficent number of SWE stations' within each zone. This argument rests on citation 22 which does not seem to be a textbook and proper scientific reference, and which does not deal with the snow topic. Moreover region C has only 3 stations in it which I doubt will inform on the spatial structure of the SWE variability. I see later that you don't use the two zones with low station density. Mention this earlier in the methods section in the same paragraph about minimum station density.

- Use of the Gini coefficient: altought this seems interesting and perhaps innovative, the use of the Gini coefficient seems quite different than what I could find with a brief litterature search and as such the way it is used in the present study should be toroughly described.  My main conecers is about the way the stations are entered in the calculation which is not explained.  How do you decide what is the 1st station chosen, the 2nd, 3rd, etc? Surely this has an important impact on the cumulative share of SWE if there is spatial autocorrelation between stations? Please describe the procedure in full so that the reader can evaluate its soundness and replicate it.

- stationnarity. An assumption of temporal stationnarity was made early in the text without mentionning if this was tested or if it is important for the subsequent analyses. What about first-order spatial stationnarity? Have you checked it and removed it using, e.g. a polynomial trend surface as is the common approach in variagram analysis?

Author Response

Question 1

The introduction, problematic and objectives must be rewritten in a more logical and coherent way. At present, it is very general and often vague. A few concepts, such as 'delineated structures', 'spatial structures', 'model' are used without explaining what they mean. Although I understand the objectives of the study, the introduction and problematic currently do not correctly place the research objectives in the context of existing knowledge about spatial variability of SWE and its measurement.

Answer 1

The importance, role and decreased number of snow survey stations are effectively demonstrated in the manuscript (page 2, lines 29-43). However, in the case of climate analysis, the survey station network (snow and meteorological site) is unevenly distributed and not representative of the spatial variability of the snow cover (lines 33-35). In addition, previous studies based on all the stations comprised in the snow survey station network have shown a decrease in their number since the 1980s. The reduced number makes it difficult to analyze the spatial variability of snow, particularly in Quebec.

The problematic chiefly focused on the fact that studies and works highlighting the inability of the actual snow station network to represent the spatial variability of the snow cover are scarce (this is explained in lines 52-63). The works in this manuscript are based on the results of the delineation of spatial variability structures of the SWE presented in the article published by Sena et al. (2017) (rectified in the manuscript).

The concepts of spatial structures, limits and scales of observation are explained in the manuscript (page 2, line 3 and page 3, lines 37-39). Often, the spatial structure is small or not taken into account in the representativeness of the spatial variability of the snow cover the way it should. The aim of the article is to analyze the snow survey station network according to the structures of the spatial variability of the snow cover (SWE in such case) (lines 70-72). The aims of this work and the methodological approach used to achieve them are detailed on page 2, lines 78-82.

Question 2

METHODS

The description of data and methods is not sufficiently detailed or well explained.

— It is not clear if the 367 sites are all snow survey networks with similar measurement protocols.

Answer 2

Further descriptions of the data and measurement method have been added (page 3, lines 156-173). The description of the snow water equivalent data collected from the snow survey site network is detailed on page 4, lines 132-142. Snow water equivalent measurements are made following the sampling protocols established by Environment Canada/Meteorological Service of Canada. All snow sites (MDDEFP, Rio Tinto, and Hydro-QuĂ©bec) apply the same sampling protocols. References to the protocols have been added to the manuscript (page 4, line 142).

Question 3

— number of station per regional and local zone. The table on page 7 has no number and is not cited in the text. It seems to also contain inaccuracies (mean SWE vs. mean number of SWE stations?). I question the arguments provided for “having a sufficient number of SWE stations” within each zone. This argument rests on citation 22 which does not seem to be a textbook and proper scientific reference, and which does not deal with the snow topic. Moreover, region C has only 3 stations in it which I doubt will inform on the spatial structure of the SWE variability. I see later that you don't use the two zones with low station density. Mention this earlier in the methods section in the same paragraph about minimum station density.

Answer 3

Explanations were added to the manuscript to improve the understanding of Table 1 (page 7, lines 218–223 and lines 218–223). To describe a regionalized variable that extends into a geographic area and that has simple spatial structures, a variographic analysis is an appropriate tool [1]. However, sufficient samples are needed to minimize analysis-induced errors. In areas A and C, the sample size is very small and cannot be used for variographic analysis. For some homogeneous local areas in—terms of snow water equivalent—are almost entirely devoid of snow sites (page 7, lines 214-217). The bibliographic reference was updated in the manuscript (page 7, line 215).

Question 4

— Use of the Gini coefficient: although this seems interesting and perhaps innovative, the use of the Gini coefficient seems quite different from what I could find with a brief literature search and as such the way it is used in the present study should be thoroughly described. My main concern is about the way the stations are used in the calculation which is not explained. How do you decide what is the 1st station chosen, the 2nd, 3rd, etc.? Surely this has an important impact on the cumulative share of SWE if there is spatial autocorrelation between stations? Please describe the procedure in full so that the reader can evaluate its soundness and replicate it.

Answer 4.

As mentioned above, one of the aims of the manuscript is to analyze the spatial distribution of snow survey stations in the homogeneous zones delimited in terms of SWE (page 3, line 82). The Gini coefficient is applied to the cumulative share of the number of stations (Y) and the cumulative share of the annual averages of the SWE (page 7, lines 218-220). The statistical values of SWE calculated for each snow sites were specified in the manuscript (page 4, lines 152-154). In this analysis, it is the spatial distribution of snow sites in each area that was considered. The Gini coefficient interpretation based on spatial dispersion does not involve the order of attribution of the measuring sites or the correlation between the stations [2, 3].

Question 5

— Stationary. An assumption of temporal stationarity was made early in the text without mentioning if this was tested or if it is important for the subsequent analyses. What about first-order spatial stationarity? Have you checked it and removed it using, e.g. a polynomial trend surface as is the common approach in variogram analysis?

Answer 5

Snow survey observations have been collected for a period ranging from 5 to 100 years. It has been decided to use the data from the snow survey stations that have at least 10 years of measurements [4], and we assumed that the snow phenomenon was stationary throughout this period of observation. Should it be demonstrated that the observations of the annual maximum of the SWE are not stationary over time, the observation periods would be subdivided into sub-periods for which the annual maximum of the SWE could be considered stationary. For each site, the annual maximum SWE was extracted and the mean throughout the period was calculated. For this study, we assumed that the snow cover was spatially stationary over the observation period. If it is shown that observations of annual maximum SWE are not stationary over time (which is likely to be the case according to a previous study [5]), observation periods should be divided into sub-periods for which the maximum of the SWE could be considered as stationary. This basic hypothesis allowed us to explicitly delineate the boundaries of the structures of the spatial variability of the SWE.

References

1.            Emery, X. GĂ©ostatistique LinĂ©aire. In GĂ©ostatistique LinĂ©aire, GĂ©ostatistique, É.d.M.d.P.C.d., Ed. École des Mines de Paris. Centre de GĂ©ostatistique, 2001; p. 405.

2.            Biraben, J.-N.; Duhourcau, F. La mesure de la population dans l'espace. Population 1974, 29, 113-137.

3.            Canard, A.; Poinsot, D. Quelques mĂ©thodes statistiques:Typiques de l’étude des populations et des peuplements par la mĂ©thode des quadrats. 2014.

4.            Sena, N.; Chokmani, K.; Gloaguen, E.; Bernier, M. Multi-scale analysis of the spatial variability of the water equivalent of snow (EEN) on the eastern territories of Canada. Hydrological Sciences Journal-Journal des Sciences Hydrologiques 2017, 62, 359-377.

5.            Brown, R.D. Analysis of snow cover variability and change in QuĂ©bec, 1948-2005. Hydrological Processes 2010, 10.1002/hyp.7565, 26, doi:10.1002/hyp.7565.

6.            Matheron, G. Les variables rĂ©gionalisĂ©es et leur estimation: une application de la thĂ©orie des fonctions alĂ©atoires aux sciences de la nature; Masson et CIE: 1965.

Reviewer 2 Report

This manuscript conducts spatial variability analysis on a network of snow survey network stations in eastern Canada. The paper, in general is well written. I think that extensive work has been carried out concerning the spatial distribution of SWE at multiple scales and the authors fail to recognize these in the introduction.

From a scientific perspective, it is unclear how the analysis can accomplish the goals of the paper. I think that the variogram analysis is appropriate in determining the scale of variability. The Gini Index and Lorenz curve are interesting and I appreciate the novel approach. However, if I am interpreting it correctly, it really shows how homegeneous max SWE is in the region of compared stations. In other words, if all stations have the same SWE, then the slope of the line will be closer to 1:1.  But we know that SWE varies and the variograms show this as well. So it is unclear to me the benefit of the Gini index in the context that it is being applied. I think using the Gini Index for random subsamples of stations to determine new local zones to consider homogeneous SWE could be interesting. This would be assuming that stations cover the variability of the are, though. 

Furthermore, the results go in to far too much detail in the text. A table would help to summarize the results and then the text could generalize some of the values such as max, min, or mean.

In order to make the conclusion that more stations are needed for an area, how do you really know? Is there some way to see if the average SWE from the stations is correct over the entire area? It seems to me that more data are needed to draw the desired conclusions. Some more discussion of the application are also necessary.

At this time, I think that the manuscript needs significant improvement before publication including additional analyses to make more conclusive statements. 

Author Response

Reviewer 2

Question 1

This manuscript conducts spatial variability analysis on a network of snow survey network stations in eastern Canada. The paper, in general is well written. I think that extensive work has been carried out concerning the spatial distribution of SWE at multiple scales and the authors fail to recognize these in the introduction.

Answer 1

A summary of the reasoning behind the delineation of spatial variability structures of SWE at different scales is included in the manuscript (page 1, lines 35-39; page 4, lines 156-164).

Question 2

From a scientific perspective, it is unclear how the analysis can accomplish the goals of the paper. I think that the variogram analysis is appropriate in determining the scale of variability. The Gini Index and Lorenz curve are interesting and I appreciate the novel approach. However, if I am interpreting it correctly, it really shows how homogeneous max SWE is in the region of compared stations. In other words, if all stations have the same SWE, then the slope of the line will be closer to 1:1. But we know that SWE varies and the variograms show this as well. So it is unclear to me the benefit of the Gini index in the context that it is being applied. I think using the Gini Index for random subsamples of stations to determine new local zones to consider homogeneous SWE could be interesting. This would be assuming that stations cover the variability of the area. Though, furthermore, the results go into far too much detail in the text. A table would help to summarize the results and then the text could generalize some values such as max, min, or mean.

Answer 2

As mentioned in the manuscript, the aims are: 1) to assess the current snow survey network's capacity to capture the spatial variability of the mean annual maximum SWE, and 2) to analyze the spatial distribution of the snow survey stations in the delineated structures. To describe a regionalized variable that extends into a geographic area and has simple spatial structures, the variographic analysis is the appropriate tool [1]. The variogram provides the characteristics of the structure (variance, the distance from which it is stable, microstructure nor nugget effect). The SWE values are not the same in stations located in an area (this is demonstrated by variogram analysis at the regional scale).

The Gini index shows the degree of spatial distribution of the stations within the area. As explained in the manuscript, the bisector represents an even distribution of the measurement stations rather than the same value of SWE at each station (page 8, line 225). Thus, in areas where the spatial distribution of the stations is not proportional, the slope of the Lorenz curve decreases (page 11, lines 302-307). At a local scale, interpretation of the results area by area is important because the factors behind the spatial distribution of the stations are not always the same.

Question 3

In order to make the conclusion that more stations are needed for an area, how do you really know? Is there some way to see if the average SWE from the stations is correct over the entire area? It seems to me that more data are needed to draw the desired conclusions. Some more discussion of the application is also necessary.

Answer 3

It is not a question of whether more snow stations are needed in an area, but whether the snow stations found in each area are adequately distributed over space to provide data on the spatial variability of the SWE in each homogeneous area. The Gini index only gives the degree of spatial distribution of the variable of interest. One of the aims mentioned is to analyze the spatial distribution of the stations in the delimited structures (page 2, line 82).

References

1.            Emery, X. GĂ©ostatistique LinĂ©aire. In GĂ©ostatistique LinĂ©aire, GĂ©ostatistique, É.d.M.d.P.C.d., Ed. École des Mines de Paris. Centre de GĂ©ostatistique, 2001; p. 405.

2.            Biraben, J.-N.; Duhourcau, F. La mesure de la population dans l'espace. Population 1974, 29, 113-137.

3.            Canard, A.; Poinsot, D. Quelques mĂ©thodes statistiques:Typiques de l’étude des populations et des peuplements par la mĂ©thode des quadrats. 2014.

4.            Sena, N.; Chokmani, K.; Gloaguen, E.; Bernier, M. Multi-scale analysis of the spatial variability of the water equivalent of snow (EEN) on the eastern territories of Canada. Hydrological Sciences Journal-Journal des Sciences Hydrologiques 2017, 62, 359-377.

5.            Brown, R.D. Analysis of snow cover variability and change in QuĂ©bec, 1948-2005. Hydrological Processes 2010, 10.1002/hyp.7565, 26, doi:10.1002/hyp.7565.

6.            Matheron, G. Les variables rĂ©gionalisĂ©es et leur estimation: une application de la thĂ©orie des fonctions alĂ©atoires aux sciences de la nature; Masson et CIE: 1965.

Reviewer 3 Report

Review for “Critical analysis of the snow survey network according to the spatial variability of snow water equivalent (SWE) on the mainland part of Eastern Canada” by Noumonvi Yawu SENA, Karem Chokmani, Erwan Gloaguen, Monique Bernier

This study evaluated the spatial variability structures of the mean annual maximum snow water equivalent (SWE) data at both regional and local scales. To do this, they use variogram to characterize the spatial structures and use Gini coefficient to characterize the spatial distributions of snow survey stations. They conclude that at regional scale, the snow survey stations are evenly distributed within the spatial structures but not at local scale. Overall, this study delivers a clear message of the spatial heterogeneity of snow survey stations. However, I find the implications of these results are not immediately clear to the readers. I recommend the authors add more discussions on how their results can help remote sensing applications and inform various stake holders in using these snow survey data to do decision making. In addition, I have following comments that hopefully can help improve this manuscript.

·      Figure 1. What does the gray scale in the inset plot mean?

·      Line 153 and 158, are you referring to Figure 2a and 2b?

·      Figure 2. Although detailed in Sena et al., it is necessary to summarize the reasoning of the delineation of regions and subregions in Figure 2 in this manuscript. What does the plus and minus sign in a mean?

·      Line 255. What is the actual value of the mean annual maximum SWE of the Piedmont stations? Can you please annotate the nugget effect in Figure 4. It is hard to tell from thee y intercept in Figure 4.

·      There is no section 4 in the manuscript. The authors should add a discussion section in this manuscript.

Author Response

Reviewer 3

Review for “Critical analysis of the snow survey network according to the spatial variability of snow water equivalent (SWE) on the mainland part of Eastern Canada” by Noumonvi Yawu SENA, Karem Chokmani, Erwan Gloaguen, Monique Bernier

Question 1

This study evaluated the spatial variability structures of the mean annual maximum snow water equivalent (SWE) data at both regional and local scales. To do this, they use variogram to characterize the spatial structures and use Gini coefficient to characterize the spatial distributions of snow survey stations. They conclude that at the regional scale, the snow survey stations are evenly distributed within the spatial structures but not at the local scale. Overall, this study delivers a clear message of the spatial heterogeneity of snow survey stations. However, I find the implications of these results are not immediately clear to the readers. I recommend the authors add more discussions on how their results can help remote sensing applications and inform various stakeholders in using these snow survey data to do decision-making. In addition, I have following comments that hopefully can help improve this manuscript.

· Figure 1. What does the gray scale in the inset plot mean?

Answer 1

Comments have been added about the contribution of the manuscript for using in situ data in the validation of snowfall spatialization algorithms (page 21, lines 260 and 453). Necessary corrections have been made in Figure 1.

Question 2

· Line 153 and 158; are you referring to Figure 2a and 2 b?

Answer 2

The necessary corrections have been added (page 4, lines 153-158).

Question 3

· Figure 2. Although detailed in Sena et al., it is necessary to summarize the reasoning of the delineation of regions and subregions in Figure 2 in this manuscript. What does the plus and minus sign in a mean?

Answer 3

A summary of the reasoning behind the delineation of regions and subregions was added in Figure 2 (page 4, line 151-166).

Question 4

· Line 255. What is the actual value of the mean annual maximum SWE of the Piedmont stations? Can you please annotate the nugget effect in Figure 4? It is hard to tell from thee y intercept in Figure 4.

Answer 4

The represented structure obtained by the variogram analysis of SWE is a geostatistical analysis based on the regionalized variables theory and derived from the random functions [6]. The variance of the SWE does not depend on the coordinates of the snow stations but on the distance (h) between them.

In Figure 4 (zone B) for example, the structure of the SWE shows that starting at 600 km, the variance (y (hB)) of the SWE reaches a sill and remains stationary. Microstructures (nugget effect) of variance of 200 mm 2 are observed under a distance of 5 km (page 8, lines 274-282) such precision for the other zones were added in the manuscript (page 8, lines 257 and 276).

Question 5

· There is no section 4 in the manuscript. The authors should add a discussion section in this manuscript.

Answer 5

The method of combining the conclusion and the discussion is applied in this manuscript. Additional discussion elements have also been added (page 20, lines 453-460 and lines 466-469).

References

1.            Emery, X. GĂ©ostatistique LinĂ©aire. In GĂ©ostatistique LinĂ©aire, GĂ©ostatistique, É.d.M.d.P.C.d., Ed. École des Mines de Paris. Centre de GĂ©ostatistique, 2001; p. 405.

2.            Biraben, J.-N.; Duhourcau, F. La mesure de la population dans l'espace. Population 1974, 29, 113-137.

3.            Canard, A.; Poinsot, D. Quelques mĂ©thodes statistiques:Typiques de l’étude des populations et des peuplements par la mĂ©thode des quadrats. 2014.

4.            Sena, N.; Chokmani, K.; Gloaguen, E.; Bernier, M. Multi-scale analysis of the spatial variability of the water equivalent of snow (EEN) on the eastern territories of Canada. Hydrological Sciences Journal-Journal des Sciences Hydrologiques 2017, 62, 359-377.

5.            Brown, R.D. Analysis of snow cover variability and change in QuĂ©bec, 1948-2005. Hydrological Processes 2010, 10.1002/hyp.7565, 26, doi:10.1002/hyp.7565.

6.            Matheron, G. Les variables rĂ©gionalisĂ©es et leur estimation: une application de la thĂ©orie des fonctions alĂ©atoires aux sciences de la nature; Masson et CIE: 1965.

Round 2

Reviewer 1 Report

The new version of this manuscript, while having improved some of the minor comments expressed in my revision and that of other reviewers, has failed to properly and explicitly  address the main concerns expressed in my first review. The manuscript is also of questionable quality in terms of presentation and supporting references, the authors will find several comments in the commented manuscript to this effect. I still believe the topic to be of interest, and it may be that the results are after all valid and worth publishing, but the justification for the methods as given in the present version of the manuscript do not allow the reader to understand, and perhaps replicate, what was done. The response given to two of my main concerns have not been included in the main text, so that the readers will inherently be faced with the same interrogations, which I deem to be unresolved for now. As said, the authors may be able to justify and explain better the methods used and how they allow responding the objectives, but this is not possible based on the current version.

Two of my main concerns were:

(1) the use of records of varying length to calculate a statistic (mean annual SWE maximum), and the assumption of time stationarity. In response to this comment the authors have highlighted the fact that record with length between a minimum of 10 years and maximum of 100 years were used to calculate the mean annual SWE maximum, assuming time stationarity. This in fact raised more concerns. On the one hand from their response I interpreted that stationarity was not controlled for. I also see a major problem alone in calculating the mean maximum station SWE using such different record length, and knowing in addition that the time stationarity conditions is most probably wrong for long-records.  It means that much of the spatial variability analyzed could stem from the varying record length used to calculate the mean annual SWE maximum. Or have they only used 10-year long records? This is not what I interpret from their reply.

(2) Use of Gini coefficient to characterize the network distribution. The original study objectives were: to evaluate (1) the current snow survey 80 network's capacity to capture the spatial variability of the mean annual maximum SWE; and (2) the spatial distribution of the snow stations in the delineated structures is analyzed. I am not convinced that the use of the Gini coefficient has contributed to these objectives, at least not in the way it is described in the current manuscript. In my first review I highlighted the lack of details on the use and significance of the Gini coefficient in the context of the study. I saw it as a potentially novel approach but unfortunately I find it use and description to be still unsubstantiated in the current manuscript version. The authors must clearly explain how spatial clustering of station, vs. spatial variability in SWE, affect the gini coefficient. You could have two networks with the same spatial distribution and same number of station, but with different amplitude of spatial variability in SWE (as captured for example by the spatial standard deviation). These two situation would give you two different Gini coefficients reflecting the different spatial variability, independent of the network configuration. So what are we left to conclude when looking at the Gini coefficients? That one areas has more variable SWE than another? If the maxSWE distribution is gaussian and well distributed the GI will be low. If the SWEmax distribution is skewed and the network still well distributed you will get a higher GI (after all this is what the GI measures...). Hows does the network configuration affect the GI? 

The question of interest seems to be whether the observed network configuration is able to capture the mean SWE correctly, and this depends on the spatial degree of freedom (which you can estimate from your semivariogram range parameter and the area of interest). Your analysis cannot respond to this question. Are there enough stations and are they distributed in a way to allow capturing the dominant scale of variability? If the Gini coefficient does allow answering this, then you need a much better description of it in the methods, and on how to interpret the results.

Author Response

Reviewer 1

The new version of this manuscript, while having improved some of the minor comments expressed in my revision and that of other reviewers, has failed to properly and explicitly address the main concerns expressed in my first review. The manuscript is also of questionable quality in terms of presentation and supporting references, the authors will find several comments in the commented manuscript to this effect. I still believe the topic to be of interest, and it may be that the results are after all valid and worth publishing, but the justification for the methods as given in the present version of the manuscript do not allow the reader to understand, and perhaps replicate, what was done. The response given to two of my main concerns have not been included in the main text, so that the readers will inherently be faced with the same interrogations, which I deem to be unresolved for now. As said, the authors may be able to justify and explain better the methods used and how they allow responding the objectives, but this is not possible based on the current version.

Two of my main concerns were:

Question 1

(1) the use of records of varying length to calculate a statistic (mean annual SWE maximum), and the assumption of time stationarity. In response to this comment the authors have highlighted the fact that record with length between a minimum of 10 years and maximum of 100 years were used to calculate the mean annual SWE maximum, assuming time stationarity. This in fact raised more concerns. On the one hand from their response I interpreted that stationarity was not controlled for. I also see a major problem alone in calculating the mean maximum station SWE using such different record length, and knowing in addition that the time stationarity conditions is most probably wrong for long-records. It means that much of the spatial variability analyzed could stem from the varying record length used to calculate the mean annual SWE maximum. Or have they only used 10-year long records? This is not what I interpret from their reply.

Reponse 1

The snow survey sites of the study area consist of 426 stations with different measurement periods. For example, Shawinigan snow survey site has a measurement period of 83 years old, La Tuque has 76-year old and the Mine Madeleine site has 7 years old. According to the studies of snow cover variability, 10-year cyclical atmospheric and oceanic events (solar cycle, El Niño-Southern Oscillation, Nina, etc.) are likely to influence the snow accumulation (Rasmussen et al. 1999, Sobolowski and Frei 2007, Brown 2010) (Page4 line143-149). For that, we selected only all snow survey sites with an observation period of 10 years or more under the assumption that the phenomenon is stationary. For this study, we assumed that snow cover is spatially stationary in observation period. For that, the sub-period were not considered (Page 4, line15-154).This assumption is important to identify, delimite and analyze the spatial structure of SWE.

For each snow sites, we decided to work with snow survey sites having a period greater than 10 years. The annual maximum was extracted and the mean throughout the period was calculated. Thereafter, the standard deviation and the interquartile range were calculated.

In the article "Analyse multi-échelles de la variabilité spatiale de l’équivalent en eau de la neige (EEN) sur le territoire de l’Est du Canada / Multi-scale analysis of the spatial variability of the water equivalent of snow (SWE) on the eastern territories of Canada"[1], the stationary

2

assumption is more explained. In this study, it was assumed that the snow phenomenon is stationary over the entire observation period. If it is shown that the annual maximum observations of the SWE are not stationary over time (which is likely the case based on Brown 2010), the results of the spatial segmentation must be resumed. In this case, the observation periods should be subdivided for which the maximum of the SWE could be considered as stationary. Subsequently, the spatial segmentation methodology proposed in this article must be repeated. It would then be interesting to analyze the evolution of the limits of the structures according to the stationarity of the snow. This study lays the foundation for identifying the limits of spatial structures under the hypothesis of the stationarity of the snow phenomenon during the observation period.

Question 2

(2) Use of Gini coefficient to characterize the network distribution. The original study objectives were: to evaluate (1) the current snow survey 80 network's capacity to capture the spatial variability of the mean annual maximum SWE; and (2) the spatial distribution of the snow stations in the delineated structures is analyzed. I am not convinced that the use of the Gini coefficient has contributed to these objectives, at least not in the way it is described in the current manuscript. In my first review I highlighted the lack of details on the use and significance of the Gini coefficient in the context of the study. I saw it as a potentially novel approach but unfortunately I find it use and description to be still unsubstantiated in the current manuscript version. The authors must clearly explain how spatial clustering of station, vs. spatial variability in SWE, affect the gini coefficient. You could have two networks with the same spatial distribution and same number of station, but with different amplitude of spatial variability in SWE (as captured for example by the spatial standard deviation). These two situation would give you two different Gini coefficients reflecting the different spatial variability, independent of the network configuration. So what are we left to conclude when looking at the Gini coefficients? That one areas has more variable SWE than another? If the maxSWE distribution is gaussian and well distributed the GI will be low. If the SWEmax distribution is skewed and the network still well distributed you will get a higher GI (after all this is what the GI measures...). Hows does the network configuration affect the GI?

Reponse 2

The main objective of the present study is to conduct a critical and objective analysis of the existing snow survey network, based on delineated spatial structures of the mean annual maximum SWE in eastern Canada, as presented in SĂ©na, N.Y (Page 2, line 78-80)

More specifically, 1) the current snow survey network's capacity to capture the spatial variability of the mean annual maximum SWE is evaluated and

2) the spatial distribution of the snow stations in the delineated structures is analyzed. (Page 3, line 82)

Two approaches are adopted to achieve these objectives (Page3, line83-84)

In the first part, the spatial variability of the mean annual maximum SWE will be assessed using a variogram analysis in geographical areas with homogeneous spatial structures (in terms of mean annual maximum SWE).

3

In the second part, the Lorenz index will be used to characterize and measure the spatiale distribution type of the stations in the delineated spatial structure.

In the manuscript, the Gini coefficient is a statistical measure to account only for the unequal spatial distribution of station in each structure delineated (Necessary correction have been made (Page 7-9, line 239-243). Inequality is measured using a synthetic indicator that summarizes the dispersion of the spatial distribution of station and can be represented as a graph such as the Lorenz curve. The Gini index is an indicator of the type of normative measure (Page 8, line 241-243).

It is not a question of assessing SWE spatial variability versus the spatial clustering of stations by the Gini coefficient. This aim is not study in this manuscript.

Necessary corrections and information have been made in manuscript (page 6-7, line 238-244)

Question 3

The question of interest seems to be whether the observed network configuration is able to capture the mean SWE correctly, and this depends on the spatial degree of freedom (which you can estimate from your semivariogram range parameter and the area of interest). Your analysis cannot respond to this question. Are there enough stations and are they distributed in a way to allow capturing the dominant scale of variability? If the Gini coefficient does allow answering this, then you need a much better description of it in the methods, and on how to interpret the results.

Reponse 3

At the regional scale, six homogeneous geographic areas in terms of SWE are delimited. Variographic analysis is carried out in four areas with sufficient number of stations (page 7, line 224-225). At the local scale, variographic analysis can not be applied because the number of stations is insufficient (page 7, line 224-225).

The question is whether the stations in each zone can provide the SWE's variability characteristics by variographic analysis. The Gini coefficient only allows to give indications on the distribution of the stations and not their capacity to analyze the SWE's spatial variability

Reference

1. Sena, N., et al., Multi-scale analysis of the spatial variability of the water equivalent of snow (EEN) on the eastern territories of Canada. Hydrological Sciences Journal, 2017. 62(3): p. 359-377.

Reviewer 2 Report

Thank you for revising the manuscript. However, it does not appear that much has occurred to address my previous comments. I appreciate the direct responses to my concerns, but perhaps the text in the manuscript could be revised for this as well. 

Per my previous comment about the extensive work previously done in the spatial variability of snow. More is still needed to address this. I think it would be good to recognize the extensive work by some of the following: Bloschl, Pomeroy, Mote, Elder, Williams, Nolin, Jonas, Winstral, Lopez-Moreno, and many others that are not included.

It also seems that one of my comments was not interpreted correctly and I apologize for not presenting it more clearly. My comment (Question 2 in the response) was not about the max SWE being the same, But it was rather the interpretation of the data. I think that the results could be better presented to highlight the value of the analysis. The text is quite heavy and could be shortened to only contain a summary and the results be summarized in a table. 

Concerning "Question 3", SWE is going to vary and so there is a fundamental assumption in this analysis that each station is representative of an area. What evidence do you have that homogeneous zones are actually homogeneous (i.e. in between stations)? Or that a station is capturing a representative measurement for the much larger area? 

Furthermore, Webster and Oliver (2007) state that 150 data points are needed for a stable and accurate variogram. The most that you have is 128 in a region, so how do you know that you are capturing the true variability? This is the fundamental concern that I have with the work and this needs to be addressed: How do you know the variability of SWE is being captured with so few data points for such a large study domain?

Perhaps the terminology used could help clarify things for the reader. For example, if you look at table 1, the term "homogenous zone" is used to describe two very different scales. Perhaps these could be re-written to make it clear exactly what scale is meant. This may be one of the reasons that I was having difficulty following some of the manuscript.

Webster and Oliver, 2007. Geostatistics for Environmental Scientists. DOI: 10.1002/9780470517277

Author Response

Reviewer 2

Thank you for revising the manuscript. However, it does not appear that much has occurred to address my previous comments. I appreciate the direct responses to my concerns, but perhaps the text in the manuscript could be revised for this as well.

Question 1

Per my previous comment about the extensive work previously done in the spatial variability of snow. More is still needed to address this. I think it would be good to recognize the extensive work by some of the following: Bloschl, Pomeroy, Mote, Elder, Williams, Nolin, Jonas, Winstral, Lopez-Moreno, and many others that are not included.

Reponse 1

This manuscript is based on the analyze results of the spatial variability of SWE obtained by SĂ©na et al. (page 2, line 78-80). This analysis of the spatial variability of SWE is analyzed on two observation scales (10km x 10km and 300m x 300m) where the extensive work of the various authors was discussed in this analysis [1] (For more details see SĂ©na et al (2017)). This manuscript takes a critical look at the network of the measuring stations in each of the delineated structures of the spatial variability of SWE at two observation scales.

Question 2

It also seems that one of my comments was not interpreted correctly and I apologize for not presenting it more clearly. My comment (Question 2 in the response) was not about the max SWE being the same, But it was rather the interpretation of the data. I think that the results could be better presented to highlight the value of the analysis. The text is quite heavy and could be shortened to only contain a summary and the results are summarized in a table.

Reponse 2

In the local units, some measuring snow sites are located along parks, along roads, in economic interest areas and others at low altitudes. Very often, the measuring snow sites spatial distribution, is guided by geographic factors (mountain presence, urban agglomerations and hydroelectric industries) as local units of Churchill Falls (unit 1 of zone B), around the Laurier Mountains (unit 10 of zone B) and low altitudes (unit 1 of zone D) for example. These different distributions should be detailed in order to guide readers who have limited knowledge about the study area. We chose this more detailed outcome option to help understand the spatial distribution of measuring snow sites at each unit scale. A table would give a smaller approach to the discussion, especially since the high concentrations of measurement stations are located in different areas of economic interest distributed differently in delimited units and zones. Interpretation of the results in area or units is important because the factors behind the spatial distribution of the measuring snow sites are not always the same.

Question 3

Concerning "Question 3", SWE is going to vary and so there is a fundamental assumption in this analysis that each station is representative of an area. What evidence do you have that

homogeneous zones are actually homogeneous (i.e. in between stations)? Or that a station is capturing a representative measurement for the much larger area?

Reponse 3

The results of Sena et al. (2017) studies showed at the regional scale (10km x 10km), six homogeneous spatial structures in terms of the SWE. At the local scale (300m x 300m) several local units are identified in each zone. The multi-scale analysis of the spatial variability of SWE using a functional approach allowed the identification, characterization and delineation of the different spatial structures of SWE variability.

The results of the segmentation have been validated using the non-parametric Kruskal-Wallis statistical test applied to the adjacent SWE of each pair of geographic data. At the regional scale, the spatial segmentation has identified six (6) geographical areas distinguished by the disposal of estates of the relief. At the local scale, the spatial segmentation highlighted the role of physiographic factors (slope, curvature, land cover etc.) in the spatial variability of snow cover (Sena et al. (2017).

The aim of this manuscript is to carry out a critical analysis of the network of stations by evaluating its capacity to evaluate the characteristics of the structure of the spatial variability of the SWE and to analyze its spatial distribution in each delimited zone. At the local scale, the reduced number or absence of stations is a limit of assessment of the characteristics of the spatial variability of the SWE

Question 4

Furthermore, Webster and Oliver (2007) state that 150 data points are needed for a stable and accurate variogram. The most that you have is 128 in a region, so how do you know that you are capturing the true variability? This is the fundamental concern that I have with the work and this needs to be addressed: How do you know the variability of SWE is being captured with so few data points for such a large study domain?

Reponse 4

Webster and Oliver (2007) recommend 100 sampling point and ideally 150 and the results show clearly that sample variograms from only 25 and 49 data have wide confidence intervals, and are therefore imprecise (Page 122 on Webster and oliver 2007). In this manuscript, according the table 1, at regional scale, homogeneous zones (A and C) have respectly 8 and 3 sampling sites. Variogram analysis is not applying in this zones. According others studies minimum 50 sampling, we applied variographic analysis on 103 and 50 sites which corresponds to 5253 and 1225 pairs of variograms (n * (n-1) / 2) points (zone B and Zone D respectly) [2,3].The lack of data (less than 10 per unit) noted in homogeneous units at the local scale (300m x 300m) does not allow to characterize the real variability of SWE at this scale observation by variogram.

Question 5

Perhaps the terminology used could help clarify things for the reader. For example, if you look at table 1, the term "homogenous zone" is used to describe two very different scales. Perhaps these could be re-written to make it clear exactly what scale is meant. This may be one of the reasons that I was having difficulty following some of the manuscript.

Reponse 5

The spatial variability of the SWE analysis was made at two scales observation: regional scale (10km X10km) and the local scale (300mx300m) (table1). The spatial segmentation has been validated using the non-parametric test Kruskal-Wallis applied to the SWE data of each pair of geographic areas adjacent [1]. At the regional scale, the spatial segmentation has identified six geographical areas (table1) distinguished by their position related to the different modes of atmospheric circulation variability and the disposal of the relief. At the local scale, homogeneous geographical units in terms of the SWE are segmented. In this scale, the spatial segmentation highlighted the role of the slope, curvature etc. in the spatial variability of SWE.

1. Sena, N.; Chokmani, K.; Gloaguen, E.; Bernier, M. Multi-scale analysis of the spatial variability of the water equivalent of snow (EEN) on the eastern territories of Canada. Hydrological Sciences Journal-Journal des Sciences Hydrologiques 2017, 62, 359-377.

2. Coburn, T.C.; Yarus, J.M.; Chambers, R.L. Stochastic modeling and geostatistics: principles, methods, and case studies, vol. II, AAPG computer applications in geology 5; AAPG: 2005; Vol. 5.

3. Smith, T.E. Notebook on spatial data analysis. Lecture Note 2016.

Round 3

Reviewer 2 Report

The revisions that were made were rather minor and the authors' comments somewhat dismissive of my previous concerns. I believe that addressing my previous comments would greatly improve the manuscript to be more readable and increase the impact on the snow hydrology community. It is my recommendation that the authors' take this into consideration for further major revisions. 

More detailed comments are below:

Referring to my previous comments in the last 2 reviews, there is a lot of work that has been conducted in the spatial variability of snow. While I understand that this work is building upon the previous work conducted by the lead author, there is still a lot of work that needs to be mentioned in the background material.

Again, it appears that my comment has been misinterpreted. I think more discussion needs to be directed towards the number of samples. Yes, it is OK to use 50 points for variogram analysis, but this increases the uncertainty. How does this impact the results and implications? This is never discussed.

Furthermore, the uncertainty is never even mentioned. For a statistical analysis to not include any uncertainty discussion is a major shortcoming, in my opinion.

Additionally, please see my last two reviews and consider the comments as constructive. It is my intention to not be critical and bring about defensiveness, but rather provide constructive feedback to better highlight the value of the work.

Author Response

Sir, 

I would like to thank you again for the very important remarks you mentioned for improving this work. In science, all remarks are intended to lead authors to improve the results.

Thank you

We added basic information in other parts of the document:

introduction; p.2 line44-49 and line 91-95

paragraph 2.1 Territory of Study (p.3 line 107-113)

Section 2.4.2 .Study of the spatial distribution of snow survey stations (p 9, line 269-275)

Results paragraph, all

Table 2: Parameters of the variograms of the B, D, E, F and regional scales

p.11, line 333-340)

All results are more explained and in conclusion, the impact of the small size of the number of stations is mentioned in the confidence intervals of results. The perception studies are mentioned
